# Why Do Better Loss Functions
# Lead to Less Transferable Features?

**Simon Kornblith**[1][*]**, Ting Chen**[1]**, Honglak Lee**[2][†]**, Mohammad Norouzi**[1]
[1]Google Research, Toronto  [2]University of Michigan

## Abstract

Previous work has proposed many new loss functions and regularizers that improve test accuracy on image classification tasks. However, it is not clear whether these loss functions learn better representations for downstream tasks. This paper studies how the choice of training objective affects the transferability of the hidden representations of convolutional neural networks trained on ImageNet. We show that many objectives lead to statistically significant improvements in ImageNet accuracy over vanilla softmax cross-entropy, but the resulting fixed feature extractors transfer substantially worse to downstream tasks, and the choice of loss has little effect when networks are fully fine-tuned on the new tasks. Using centered kernel alignment to measure similarity between hidden representations of networks, we find that differences among loss functions are apparent only in the last few layers of the network. We delve deeper into representations of the penultimate layer, finding that different objectives and hyperparameter combinations lead to dramatically different levels of class separation. Representations with higher class separation obtain higher accuracy on the original task, but their features are less useful for downstream tasks. Our results suggest there exists a trade-off between learning invariant features for the original task and features relevant for transfer tasks.

## 1 Introduction

Features learned by deep neural networks on ImageNet transfer effectively to a wide range of computer vision tasks [23, 60, 34]. These networks are often pretrained using vanilla softmax cross-entropy [10, 11], but recent work reports that other loss functions such as label smoothing [66] and sigmoid cross-entropy [8] outperform softmax cross-entropy on ImageNet. Although a previous investigation suggested that label smoothing and dropout can hurt transfer learning [34], the vast majority of work studying alternatives to softmax cross-entropy has restricted its empirical investigation to the accuracy of the trained networks on the original dataset. While it is valuable to study the impact of objective functions on classification accuracy, improving transfer accuracy of learned representations can have a more significant practical impact for many applications.

This paper takes a comparative approach to understand the effects of different training objectives for image classification through the lens of neural network representations and their transferability. We carefully tune hyperparameters of each loss function and confirm that several loss functions outperform softmax cross-entropy by a statistically significant margin on ImageNet. However, we find that these improvements do not transfer to other tasks (see Table 1 and Figure 1). We delve deeper and explain these empirical findings in terms of effects of different objectives on neural network representations. Our key findings are as follows:

- We analyze the performance of 9 different objectives on ImageNet and in transfer settings. Although many loss functions and regularizers lead to statistically significant improvements over vanilla

---

[*]Correspondence to: `skornblith@google.com`

[†]Work performed while at Google.

35th Conference on Neural Information Processing Systems (NeurIPS 2021).

softmax cross-entropy on ImageNet, these gains do not transfer. These alternative loss functions produce fixed feature extractors that transfer substantially worse to other tasks, in terms of both linear and k-nearest neighbors classification accuracy, and provide no benefit when representations are fully fine-tuned.

- The choice of objective primarily affects representations in network layers close to the output. Centered kernel alignment (CKA) reveals large differences in representations of the last few layers of the network, whereas earlier layers are similar regardless of which training objective is used. This helps explain why the choice of objective has little impact on fine-tuning transfer accuracy.

- All objectives that improve accuracy over softmax cross-entropy also lead to greater separation between representations of different classes in the penultimate layer features. These alternative objectives appear to collapse within-class variability in representations, which accounts for both the improvement in accuracy on the original task and the reduction in the quality of the features on downstream tasks.

## 2 Loss Functions and Output Layer Regularizers

We investigate 9 loss functions and output layer regularizers. Let $z \in \mathbb{R}^K$ denote the network's output ("logit") vector, and let $t \in \{0, 1\}^K$ denote a one-hot vector of targets. Let $x \in \mathbb{R}^M$ denote the vector of penultimate layer activations, which gives rise to the output vector as $z = Wx + b$, where $W \in \mathbb{R}^{K \times M}$ is the matrix of final layer weights, and $b$ is a vector of biases.

All investigated loss functions include a term that encourages $z$ to have a high dot product with $t$. To avoid solutions that make this dot product large simply by increasing the scale of $z$, these loss functions must also include one or more contractive terms and/or normalize $z$. Many "regularizers" correspond to additional contractive terms added to the loss, so we do not draw a firm distinction between loss functions and regularizers. We describe each objective in detail below, and provide hyperparameters in Appendix A.1.

**Softmax cross-entropy** [10, 11] is the de facto loss function for multi-class classification in deep learning:

$$\mathcal{L}_{\text{softmax}}(z, t) = -\sum_{k=1}^{K} t_k \log \left( \frac{e^{z_k}}{\sum_{j=1}^{K} e^{z_j}} \right) = -\sum_{k=1}^{K} t_k z_k + \log \sum_{k=1}^{K} e^{z_k}. \tag{1}$$

The loss consists of a term that maximizes the dot product between the logits and targets, as well as a contractive term that minimizes the LogSumExp of the logits.

**Label smoothing** [66] "smooths" the targets for softmax cross-entropy. The new targets are given by mixing the original targets with a uniform distribution over all labels, $t' = t \times (1 - \alpha) + \alpha/K$, where $\alpha$ determines the weighting of the original and uniform targets. In order to maintain the same scale for the gradient with respect to the positive logit, in our experiments, we scale the label smoothing loss by $1/(1 - \alpha)$. The resulting loss is:

$$\mathcal{L}_{\text{smooth}}(z, t; \alpha) = -\frac{1}{1 - \alpha} \sum_{k=1}^{K} \left( (1 - \alpha)t_k + \frac{\alpha}{K} \right) \log \left( \frac{e^{z_k}}{\sum_{j=1}^{K} e^{z_j}} \right) \tag{2}$$

Müller et al. [43] previously showed that label smoothing improves calibration and encourages class centroids to lie at the vertices of a regular simplex.

**Dropout on penultimate layer**: Dropout [64] is among the most prominent regularizers in the deep learning literature. We consider dropout applied to the penultimate layer of the neural network, i.e., when inputs to the final layer are randomly kept with some probability $\rho$. When employing dropout, we replace the penultimate layer activations $x$ with $\tilde{x} = x \odot \xi/\rho$ where $\xi_i \sim \text{Bernoulli}(\rho)$. Writing the dropped out logits as $\tilde{z} = W\tilde{x} + b$, the dropout loss is:

$$\mathcal{L}_{\text{dropout}}(W, b, x, t; p) = \mathbb{E}_{\xi} \left[ \mathcal{L}_{\text{softmax}}(\tilde{z}, t) \right] = -\sum_{k=1}^{K} t_k z_k + \mathbb{E}_{\xi} \left[ \log \sum_{k=1}^{K} e^{\tilde{z}_k} \right]. \tag{3}$$

Dropout produces both implicit regularization, by introducing noise into the optimization process [75], and explicit regularization, by changing the parameters that minimize the loss [69].

**Extra final layer $L^2$ regularization**: It is common to place the same $L^2$ regularization on the final layer as elsewhere in the network. However, we find that applying greater $L^2$ regularization to the final layer can improve performance. The corresponding loss is:

$$\mathcal{L}_{\text{extra\_l2}}(\boldsymbol{W}, \boldsymbol{z}, \boldsymbol{t}; \lambda_{\text{final}}) = \mathcal{L}_{\text{softmax}}(\boldsymbol{z}, \boldsymbol{t}) + \lambda_{\text{final}} \|\boldsymbol{W}\|_{\text{F}}^2. \quad (4)$$

In architectures with batch normalization, adding additional $L^2$ regularization has no explicit regularizing effect if the learnable scale ($\gamma$) parameters are unregularized, but it still exerts an implicit regularizing effect by altering optimization.

**Logit penalty**: Whereas label smoothing encourages logits not to be too negative, and dropout imposes a penalty on the logits that depends on the covariance of the weights, an alternative possibility is simply to explicitly constrain logits to be small in $L^2$ norm:

$$\mathcal{L}_{\text{logit\_penalty}}(\boldsymbol{z}, \boldsymbol{t}; \beta) = \mathcal{L}_{\text{softmax}}(\boldsymbol{z}, \boldsymbol{t}) + \beta \|\boldsymbol{z}\|^2. \quad (5)$$

Dauphin and Cubuk [19] showed that this regularizer yields accuracy improvements comparable to dropout.

**Logit normalization**: We consider the use of $L^2$ *normalization*, rather than regularization, of the logits. Because the entropy of the output of the softmax function depends on the scale of the logits, which is lost after normalization, we introduce an additional temperature parameter $\tau$ that controls the magnitude of the logit vector, and thus, indirectly, the minimum entropy of the output distribution:

$$\mathcal{L}_{\text{logit\_norm}}(\boldsymbol{z}, \boldsymbol{t}; \tau) = \mathcal{L}_{\text{softmax}}(\boldsymbol{z}/(\tau \|\boldsymbol{z}\|), \boldsymbol{t}) = -\frac{1}{\tau \|\boldsymbol{z}\|} \sum_{k=1}^{K} t_k z_k + \log \sum_{k=1}^{K} e^{z_k/(\tau \|\boldsymbol{z}\|)}. \quad (6)$$

**Cosine softmax**: We additionally consider $L^2$ normalization of both the penultimate layer features and the final layer weights corresponding to each class. This loss is equivalent to softmax cross-entropy loss if the logits are given by cosine similarity $\text{sim}(\boldsymbol{x}, \boldsymbol{y}) = \boldsymbol{x}^\mathsf{T} \boldsymbol{y}/(\|\boldsymbol{x}\|\|\boldsymbol{y}\|)$ between the weight vector and the penultimate layer plus a per-class bias:

$$\mathcal{L}_{\cos}(\boldsymbol{W}, \boldsymbol{b}, \boldsymbol{x}, \boldsymbol{t}; \tau) = -\sum_{k=1}^{K} t_k \left(\text{sim}(\boldsymbol{W}_{k,:}, \boldsymbol{x})/\tau + b_k\right) + \log \sum_{k=1}^{K} e^{\text{sim}(\boldsymbol{W}_{k,:}, \boldsymbol{x})/\tau + b_k}, \quad (7)$$

where $\tau$ is a temperature parameter as above. Similar losses have appeared in previous literature [54, 71, 76, 72, 73, 21, 38, 15], and variants have introduced explicit additive or multiplicative margins to this loss that we do not consider here [38, 72, 73, 21]. We observe that, even without an explicit margin, manipulating the temperature has a large impact on observed class separation.

**Sigmoid cross-entropy**, also known as binary cross-entropy, is the natural analog to softmax cross-entropy for multi-label classification problems. Although we investigate only single-label multi-class classification tasks, we train networks with sigmoid cross-entropy and evaluate accuracy by ranking the logits of the sigmoids. This approach is related to the one-versus-rest strategy for converting binary classifiers to multi-class classifiers. The sigmoid cross-entropy loss is:

$$
\begin{aligned}
\mathcal{L}_{\text{sce}}(\boldsymbol{z}, \boldsymbol{t}) &= -\sum_{k=1}^{K} \left( t_k \log \left( \frac{e^{z_k}}{e^{z_k} + 1} \right) + (1 - t_k) \log \left( 1 - \frac{e^{z_k}}{e^{z_k} + 1} \right) \right) \\
&= -\sum_{k=1}^{K} t_k z_k + \sum_{k=1}^{K} \log(e^{z_k} + 1). \quad (8)
\end{aligned}
$$

The LogSumExp term of softmax loss is replaced with the sum of the softplus-transformed logits. We initialize the biases of the logits $\boldsymbol{b}$ to $-\log(K)$ so that the initial output probabilities are approximately $1/K$. Beyer et al. [8] have previously shown that sigmoid cross-entropy loss leads to improved accuracy on ImageNet relative to softmax cross-entropy.

**Squared error**: Finally, we investigate squared error loss, as formulated by Hui and Belkin [30]:

$$\mathcal{L}_{\text{mse}}(\boldsymbol{z}, \boldsymbol{t}; \kappa, M) = \frac{1}{K} \sum_{k=1}^{K} \left( \kappa t_k (z_k - M)^2 + (1 - t_k) z_k^2 \right), \quad (9)$$

where $\kappa$ and $M$ are hyperparameters. $\kappa$ sets the strength of the loss for the correct class relative to incorrect classes, whereas $M$ controls the magnitude of the correct class target. When $\kappa = M = 1$, the loss is simply the mean squared error between $\boldsymbol{z}$ and $\boldsymbol{t}$. Like Hui and Belkin [30], we find that placing greater weight on the correct class slightly improves ImageNet accuracy.

# 3 Results

For each loss, we trained 8 ResNet-50 [29, 27] models on the ImageNet ILSVRC 2012 dataset [20, 57]. To tune loss hyperparameters and the epoch for early stopping, we performed 3 training runs per hyperparameter configuration where we held out a validation set of 50,046 ImageNet training examples. We provide further details regarding the experimental setup in Appendix A. We also confirm that our main findings hold for Inception v3 [66] models in Appendix B.

## 3.1 Better objectives improve accuracy, but do not transfer better

Table 1: **Objectives that produce higher ImageNet accuracy lead to less transferable fixed features.** "ImageNet" columns reflect accuracy of ResNet-50 models on the ImageNet validation set. "Transfer" columns reflect accuracy of $L^2$-regularized multinomial logistic regression or $k$-nearest neighbors classifiers trained to classify different transfer datasets using the fixed penultimate layer features of the ImageNet-trained networks. Numbers are averaged over 8 different pretraining runs; values not significantly different than the best are bold-faced ($p < 0.05$, $t$-test). The strength of $L^2$ regularization is selected on a holdout set, and $k$ is selected using leave-one-out cross-validation on the training set. See Appendix Table B.1 for a similar table for Inception v3 models, Appendix D.2 for linear transfer accuracy evaluated at different epochs of ImageNet pretraining, and Appendix D.3 for similar findings on the Chexpert chest X-ray dataset [31].

| | ImageNet | | Transfer | | | | | | | |
|---|---|---|---|---|---|---|---|---|---|---|
| Pretraining loss | Top-1 | Top-5 | Food | CIFAR10 | CIFAR100 | Birdsnap | SUN397 | Cars | Pets | Flowers |
| *Linear transfer:* | | | | | | | | | | |
| Softmax | 77.0 | 93.40 | **74.6** | **92.4** | **76.9** | **55.4** | **62.0** | 60.3 | 92.0 | **94.0** |
| Label smoothing | 77.6 | 93.78 | 72.7 | 91.6 | 75.2 | 53.6 | 61.6 | 54.8 | **92.9** | 91.9 |
| Sigmoid | **77.9** | 93.50 | 73.4 | 91.7 | 75.7 | 52.3 | **62.0** | 56.1 | 92.5 | 92.9 |
| More final layer $L^2$ | 77.7 | 93.79 | 70.6 | 91.0 | 73.7 | 51.5 | 60.1 | 50.3 | 92.4 | 89.8 |
| Dropout | 77.5 | 93.62 | 72.6 | 91.4 | 75.0 | 53.6 | 61.2 | 54.7 | **92.6** | 92.1 |
| Logit penalty | 77.7 | **93.83** | 68.1 | 90.2 | 72.3 | 48.1 | 59.0 | 48.3 | 92.3 | 86.6 |
| Logit normalization | **77.8** | 93.71 | 66.3 | 90.5 | 72.9 | 50.7 | 58.1 | 45.4 | 92.0 | 82.9 |
| Cosine softmax | **77.9** | 93.86 | 62.0 | 89.9 | 71.3 | 45.4 | 55.0 | 36.7 | 91.1 | 75.3 |
| Squared error | 77.2 | 92.79 | 39.8 | 82.2 | 56.3 | 21.8 | 39.9 | 15.3 | 84.7 | 46.7 |
| *K-nearest neighbors:* | | | | | | | | | | |
| Softmax | 77.0 | 93.40 | **60.9** | **88.8** | **67.4** | **38.4** | 53.0 | **28.9** | 88.8 | **83.6** |
| Label smoothing | 77.6 | 93.78 | 59.2 | 88.3 | 66.3 | **39.2** | **53.5** | 27.3 | **91.5** | 80.3 |
| Sigmoid | **77.9** | 93.50 | 58.5 | 88.2 | 66.6 | 34.4 | 52.8 | 26.7 | 90.8 | 81.5 |
| More final layer $L^2$ | 77.7 | 93.79 | 55.0 | 87.7 | 65.0 | 36.2 | 51.1 | 24.4 | 90.9 | 75.6 |
| Dropout | 77.5 | 93.62 | 58.4 | 88.2 | 66.4 | 38.1 | 52.4 | 27.8 | 91.1 | 80.1 |
| Logit penalty | 77.7 | **93.83** | 52.5 | 86.5 | 62.9 | 31.0 | 49.3 | 22.4 | 90.8 | 68.1 |
| Logit normalization | **77.8** | 93.71 | 50.9 | 86.6 | 63.1 | 35.1 | 45.8 | 24.1 | 88.2 | 63.1 |
| Cosine softmax | **77.9** | 93.86 | 45.5 | 86.0 | 61.5 | 28.8 | 41.8 | 19.0 | 87.0 | 52.7 |
| Squared error | 77.2 | 92.79 | 27.7 | 74.5 | 44.3 | 13.8 | 28.6 | 9.1 | 82.6 | 28.2 |

We found that, when properly tuned, all investigated objectives except squared error provide a statistically significant improvement over softmax cross-entropy, as shown in the left two columns of Table 1. The gains are small but meaningful, with sigmoid cross-entropy and cosine softmax both leading to an improvement of 0.9% in top-1 accuracy over the baseline. For further discussion of differences in the training curves, robustness, calibration, and predictions of these models, see Appendix C.

Although networks trained with softmax cross-entropy attain lower ImageNet top-1 accuracy than any other loss function, they nonetheless provide the most transferable features. We evaluated the transferability of the fixed features of our ImageNet-pretrained models by training linear or k-nearest neighbors (kNN) classifiers to classify 8 different natural image datasets: Food-101 [9], CIFAR-10 and CIFAR-100 [36], Birdsnap [6], SUN397 [77], Stanford Cars [35], Oxford-IIIT Pets [51], and Oxford Flowers [48]. The results of these experiments are shown in Table 1 and Figure 1. In both linear and kNN settings, representations learned with vanilla softmax cross-entropy perform best for most tasks.

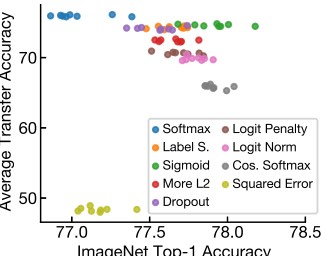

Figure 1: **Higher ImageNet accuracy is not associated with higher linear transfer accuracy.** Points represent individual training runs. See Appendix Figure D.1 for similar plots for individual datasets.

Table 2: **The training objective has little impact on the performance of fine-tuned networks.** Accuracy of fine-tuning pretrained networks on transfer datasets, averaged over 3 different pretraining initializations. We tune hyperparameters separately for each objective and dataset. Numbers not significantly different than the best are bold-faced ($p < 0.05$, $t$-test for individual datasets, two-way ANOVA for average). See Appendix A.3 for training details.

| Pretraining loss | Food | CIFAR10 | CIFAR100 | Birdsnap | SUN397 | Cars | Pets | Flowers | Avg. |
|---|---|---|---|---|---|---|---|---|---|
| Softmax | **88.2** | **96.9** | **84.1** | **76.2** | 63.4 | 91.3 | 93.1 | **96.7** | **86.2** |
| Label smoothing | **88.3** | 96.7 | 84.0 | **76.3** | 63.6 | 91.2 | **93.7** | 96.3 | 86.3 |
| Sigmoid | **88.3** | **96.9** | 83.6 | **76.2** | 63.5 | **91.8** | **93.7** | 96.3 | 86.3 |
| More final layer $L^2$ | 88.1 | **96.9** | **84.4** | 75.9 | **64.5** | 91.5 | **93.8** | 96.2 | 86.4 |
| Dropout | **88.3** | 96.7 | **84.2** | 76.5 | 63.9 | 91.2 | **93.9** | 96.3 | 86.4 |
| Logit penalty | **88.4** | **96.9** | 83.9 | **76.4** | 63.3 | 91.2 | **93.4** | **96.0** | 86.2 |
| Logit normalization | 87.9 | **96.9** | 82.9 | **76.0** | 58.3 | 91.2 | 92.7 | 95.9 | 85.2 |
| Cosine softmax | **88.3** | **96.9** | 83.2 | 75.6 | 56.9 | 91.3 | 92.5 | 95.9 | 85.1 |
| Squared error | 87.8 | **96.9** | **84.0** | 75.7 | 61.0 | 91.4 | 93.0 | 95.2 | 85.6 |

As shown in Table 2, when networks are fully fine-tuned on downstream tasks, the pretraining objective has little effect on the resulting accuracy. When averaging across all tasks, the best loss provides only a 0.2% improvement over softmax cross-entropy, which does not reach statistical significance and is much smaller than the 0.9% difference in ImageNet top-1 accuracy. Thus, using a different loss function for pretraining can improve accuracy on the pretraining task, but this improvement does not appear to transfer to downstream tasks.

## 3.2 The choice of objective primarily affects hidden representations close to the output

Our observation that "improved" objectives improve only on ImageNet and not on transfer tasks raises questions about which representations, exactly, they affect. We use two tools to investigate differences in the hidden representations of networks trained with different loss functions. First, we use centered kernel alignment [33, 17, 18] to directly measure the similarity of hidden representations across networks trained with different loss functions. Second, we measure the sparsity of the ReLU activations in each layer. Both analyses suggest that all loss functions learn similar representations throughout the majority of the network, and differences are present only in the last few ResNet blocks.

Linear centered kernel alignment (CKA) provides a way to measure similarity of neural network representations that is invariant to rotation and isotropic scaling in representation space [33, 17, 18]. Unlike other ways of measuring representational similarity between neural networks, linear CKA can identify architectural correspondences between layers of networks trained from different initializations [33], a prerequisite for comparing networks trained with different objectives. Given two matrices $X \in \mathbb{R}^{n \times p_1}$ and $Y \in \mathbb{R}^{n \times p_2}$ containing activations to the same $n$ examples, linear CKA computes the cosine similarity between the reshaped $n \times n$ covariance matrices between examples:

$$\text{CKA}_{\text{linear}}(X, Y) = \frac{\text{vec}(\text{cov}(X)) \cdot \text{vec}(\text{cov}(Y))}{\|\text{cov}(X)\|_{\text{F}}\|\text{cov}(Y)\|_{\text{F}}}. \tag{10}$$

We measured CKA between all possible pairings of ResNet blocks from 18 networks (2 different initializations for each objective). To reduce memory requirements, we used minibatch CKA [47] with minibatches of size 1500 and processed the ImageNet validation set for 10 epochs.

As shown in Figure 2, representations of the majority of network layers are highly similar regardless of loss function, but late layers differ substantially. Figure 2a shows similarity between all pairs of blocks from pairs of networks, where one network is trained with vanilla softmax cross-entropy and the other is trained with either softmax or a different loss function. The diagonals of these plots indicate the similarity between architecturally corresponding layers. For all network pairs, the diagonals are stronger than the off-diagonals, indicating that architecturally corresponding layers are more similar than non-corresponding layers. However, in the last few layers of the network, the diagonals are substantially brighter when both networks are trained with softmax than when the second network is trained with a different loss, indicating representational differences in these layers. Figure 2b shows similarity of representations among all loss functions for a subset of blocks. Consistent differences among loss functions are present only in the last third of the network, starting around block 13.

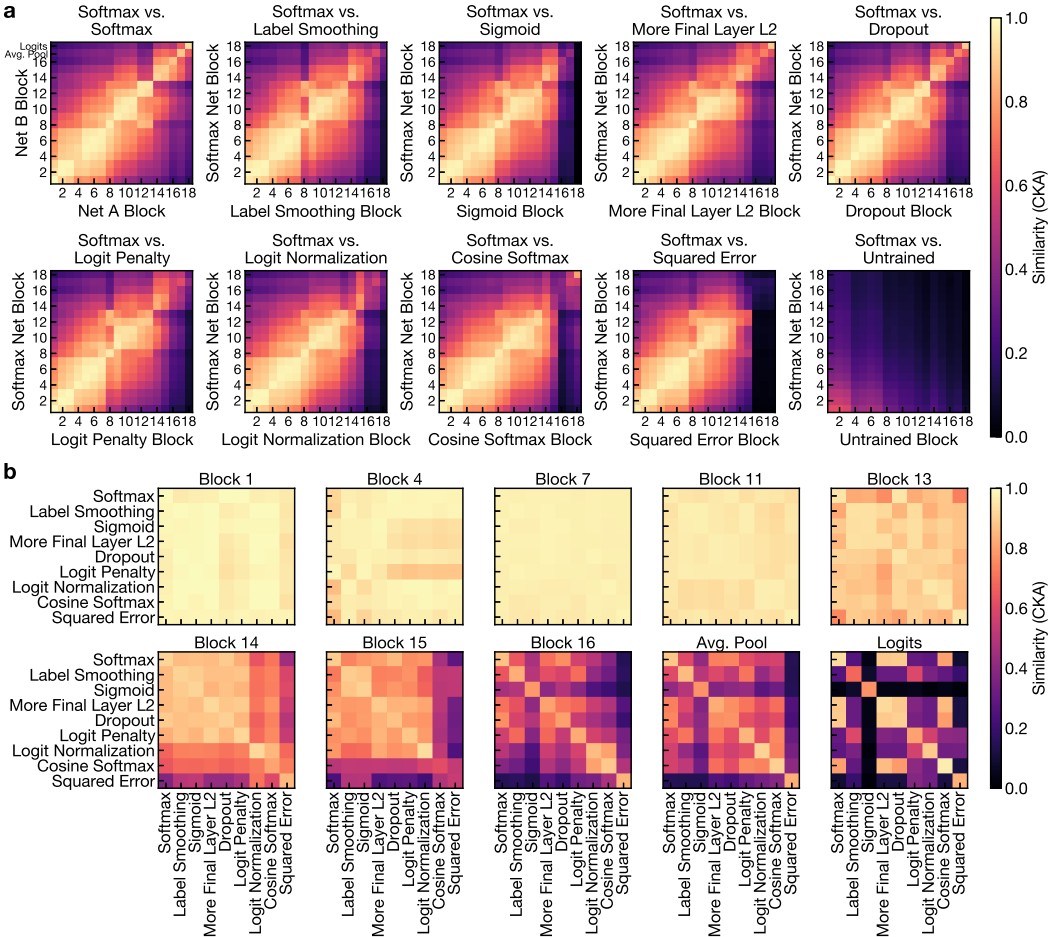

Figure 2: **The choice of loss function affects representations only in later network layers.** All plots show linear centered kernel alignment (CKA) between representations computed on the ImageNet validation set. **a**: CKA between network layers, for pairs of ResNet-50 models trained from different initializations with different losses. As controls, the top-right plot shows CKA between two networks trained with softmax, and the bottom-right plot shows CKA between a model trained with softmax loss and a model at initialization where batch norm moments have been computed on the training set. **b**: CKA between representations extracted from corresponding layers of networks trained with different loss functions. Diagonal reflects similarity of networks with the same loss function trained from different initializations. See Appendix Figure B.1 for a similar figure for Inception v3 models.

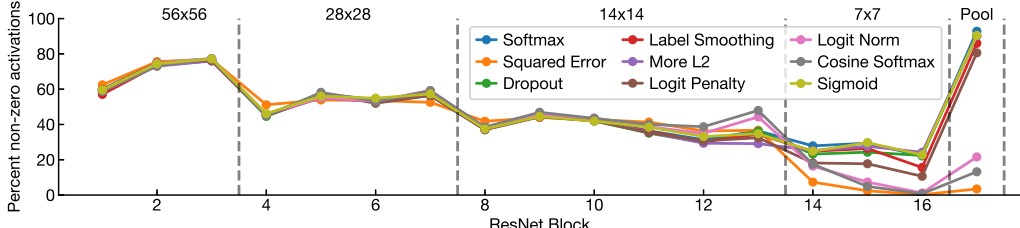

Figure 3: **Loss functions affect sparsity of later layer representations.** Plot shows the average % non-zero activations for each ResNet-50 block, after the residual connection and subsequent nonlinearity, on the ImageNet validation set. Dashed lines indicate boundaries between stages.

The sparsity of activations reveals a similar pattern of layer-wise differences between networks trained with different loss functions. Figure 3 shows the proportion of non-zero activations in different layers. In all networks, the percentage of non-zero ReLU activations decreases with depth, attaining its minimum at the last convolutional layer. In the first three ResNet stages, activation sparsity is broadly similar regardless of the loss. However, in the final stage and penultimate average pooling layer, the

Table 3: **Regularization and alternative losses improve class separation in the penultimate layer.** Results are averaged over 8 ResNet-50 models per loss on the ImageNet training set.

| Loss/regularizer | ImageNet top-1 | Class sep. ($R^2$) |
|---|---|---|
| Softmax | $77.0 \pm 0.06$ | $0.349 \pm 0.0002$ |
| Label smooth. | $77.6 \pm 0.03$ | $0.420 \pm 0.0003$ |
| Sigmoid | $77.9 \pm 0.05$ | $0.427 \pm 0.0003$ |
| Extra $L^2$ | $77.7 \pm 0.03$ | $0.572 \pm 0.0006$ |
| Dropout | $77.5 \pm 0.04$ | $0.461 \pm 0.0003$ |
| Logit penalty | $77.7 \pm 0.04$ | $0.601 \pm 0.0004$ |
| Logit norm | $77.8 \pm 0.02$ | $0.517 \pm 0.0002$ |
| Cosine softmax | $77.9 \pm 0.02$ | $0.641 \pm 0.0003$ |
| Squared error | $77.2 \pm 0.04$ | $0.845 \pm 0.0002$ |

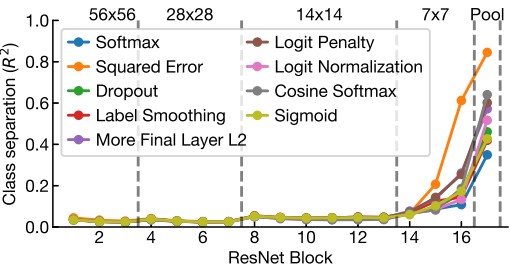

Figure 4: Class separation in different layers of ResNet-50, averaged over 8 models per loss on the ImageNet training set. For convolutional layers, we compute cosine distances by flattening the representations of examples across spatial dimensions.

degree of sparsity depends greatly on the loss. Penultimate layer representations of vanilla softmax networks are the least sparse, with 92.8% non-zero activations. Logit normalization, cosine softmax, and squared error all result in much sparser ($<25\%$ non-zero) activations.

These results immediately suggest an explanation for the limited effect of the training objective on networks' fine-tuning performance. We observe differences among objectives only in later network layers, but previous work has found that these layers change substantially during fine-tuning [79, 53, 46], an observation we replicate in Appendix D.4. Thus, the choice of training objective appears to affect parts of the network that are specific to the pretraining task, and do not transfer when the network is fine-tuned on other tasks.

### 3.3 Regularization and alternative losses increase class separation

The previous section suggests that different loss functions learn very different penultimate layer representations, even when their overall accuracy is similar. However, as shown in Appendix C.4, combining different losses and penultimate layer regularizers provides no accuracy improvements over training with only one, suggesting that they share similar mechanisms. In this section, we demonstrate that a simple property of networks' penultimate layer representations can explain their beneficial effect on accuracy relative to vanilla softmax cross-entropy, as well as their harmful effect on fixed features. Specifically, compared to vanilla softmax cross-entropy, all investigated losses cause the network to reduce the relative within-class variance in the penultimate layer representation space. This reduction in within-class variance corresponds to increased separation between classes, and is harmful to linear transfer.

The ratio of the average within-class cosine distance to the overall average cosine distance measures the dispersion of representations of examples belonging to the same class relative to the overall dispersion of embeddings. We take one minus this quantity to get a closed-form index of class separation that is between 0 and 1:

$$R^2 = 1 - \bar{d}_{\text{within}}/\bar{d}_{\text{total}} \tag{11}$$

$$\bar{d}_{\text{within}} = \sum_{k=1}^{K} \sum_{m=1}^{N_k} \sum_{n=1}^{N_k} \frac{1 - \text{sim}(\boldsymbol{x}_{k,m}, \boldsymbol{x}_{k,n})}{K N_k^2}, \quad \bar{d}_{\text{total}} = \sum_{j=1}^{K} \sum_{k=1}^{K} \sum_{m=1}^{N_j} \sum_{n=1}^{N_k} \frac{1 - \text{sim}(\boldsymbol{x}_{j,m}, \boldsymbol{x}_{k,n})}{K^2 N_j N_k}$$

where $\boldsymbol{x}_{k,m}$ is the embedding of example $m$ in class $k \in \{1, \ldots, K\}$, $N_k$ is the number of examples in class $k$, and $\text{sim}(\boldsymbol{x}, \boldsymbol{y}) = \boldsymbol{x}^\mathsf{T} \boldsymbol{y}/(\|\boldsymbol{x}\|\|\boldsymbol{y}\|)$ is cosine similarity between vectors. As we show in Appendix E.1, if the embeddings are first $L^2$ normalized, then $1 - R^2$ is the ratio of the average within-class variance to the weighted total variance, where the weights are inversely proportional to the number of examples in each class. For a balanced dataset, $R^2$ is equivalent to centered kernel alignment [17, 18] between the embeddings and the one-hot label matrix, with a cosine kernel. See Appendix E.2 for results with other distance metrics.

As shown in Table 3 and Figure 4, all regularizers and alternative loss functions we investigate produce greater class separation in penultimate layer representations as compared to vanilla softmax loss. Importantly, this increase in class separation is specific to forms of regularization that affect networks'

Figure 5: **Class separation negatively correlates with linear transfer accuracy.** **a**: Mean linear transfer accuracy across all tasks vs. class separation on ImageNet, for different loss functions. **b**: For cosine softmax loss, training at higher temperature produces greater class separation but worse linear transfer accuracy.

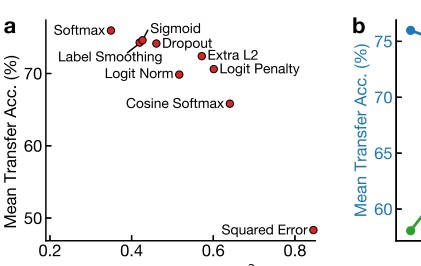

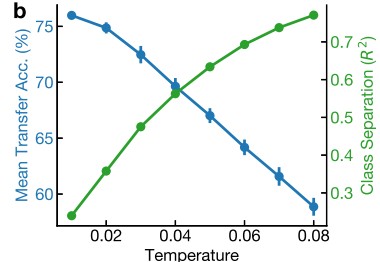

Table 4: **Temperature of cosine softmax loss controls ImageNet accuracy, class separation ($R^2$), and linear transfer accuracy.**

|  | ImageNet | | Transfer | | | | | | | |
|---|---|---|---|---|---|---|---|---|---|---|
| Temp. | Top-1 | $R^2$ | Food | CIFAR10 | CIFAR100 | Birdsnap | SUN397 | Cars | Pets | Flowers |
| 0.01 | 74.9 | 0.236 | **73.4** | **91.9** | **76.5** | **57.2** | **60.5** | **62.9** | 91.7 | **93.6** |
| 0.02 | 77.0 | 0.358 | 72.1 | 91.8 | 76.2 | 56.5 | 60.4 | 58.5 | 92.2 | 91.2 |
| 0.03 | 77.5 | 0.475 | 69.1 | 91.5 | 74.9 | 53.7 | 59.1 | 51.8 | **92.3** | 87.4 |
| 0.04 | **77.6** | 0.562 | 66.0 | 90.7 | 73.8 | 50.3 | 57.4 | 45.1 | 91.7 | 82.2 |
| 0.05 | **77.6** | 0.634 | 62.8 | 90.4 | 72.2 | 47.6 | 55.4 | 38.6 | 91.0 | 78.3 |
| 0.06 | 77.5 | 0.693 | 60.3 | 89.3 | 69.8 | 43.3 | 53.8 | 33.3 | 91.0 | 72.7 |
| 0.07 | 77.5 | 0.738 | 57.1 | 88.7 | 68.6 | 39.6 | 51.4 | 29.1 | 90.2 | 67.9 |
| 0.08 | **77.6** | **0.770** | 53.7 | 87.7 | 66.5 | 35.5 | 49.4 | 25.7 | 89.3 | 63.2 |

final layers. In Appendix E.3, we show that adding regularization through data augmentation improves accuracy without a substantial change in class separation. We further investigate the training dynamics of class separation in Appendix E.4, finding that, for softmax cross-entropy, class separation peaks early in training and then falls, whereas for other objectives, class separation either saturates or continues to rise as training progresses.

### 3.4 Greater class separation is associated with less transferable features

Although losses that improve class separation lead to higher accuracy on the ImageNet validation set, the feature extractors they learn transfer worse to other tasks. Figure 5a plots mean linear transfer accuracy versus class separation for each of the losses we investigate. We observe a significant negative correlation (Spearman's $\rho = -0.93$, $p = 0.002$). Notably, vanilla softmax cross-entropy produces the least class separation and the most transferable features, whereas squared error produces much greater class separation than other losses and leads to much lower transfer performance.

To confirm this relationship between class separation, ImageNet accuracy, and transfer, we trained models with cosine softmax with varying values of the temperature parameter $\tau$.[1] As shown in Table 4, lower temperatures yield lower top-1 accuracies and worse class separation. However, even though the lowest temperature of $\tau = 0.01$ achieved 2.7% lower accuracy on ImageNet than the best temperature, this lowest temperature gave the best features for nearly all transfer datasets. Thus, $\tau$ controls a tradeoff between the generalizability of penultimate-layer features and the accuracy on the target dataset. For $\tau \geq 0.04$, ImageNet accuracy saturates, but, as shown in Figure 5b, class separation continues to increase and transfer accuracy continues to decrease.

Is there any situation where features with greater class separation could be beneficial for a downstream task? In Figure 6, we use $L^2$-regularized logistic regression to relearn the original 1000-way ImageNet classification head from penultimate layer representations of 40,000 examples from the ImageNet validation set.[2] We find that features from networks trained with vanilla softmax loss perform *worst*, whereas features from networks with greater class separation perform substantially better. Thus, it

---

[1]Training at $\tau < 0.05$ was unstable; we scale the loss by the $\tau$ to reduce the effect of $\tau$ on the size of the gradient WRT the correct class logit. Relationships for $\tau \geq 0.05$ remain consistent without loss scaling.

[2]We use this experimental setup to avoid training linear classifiers on examples that were also seen during pretraining. However, results are qualitatively similar if we train the linear classifier on a 50,046 example subset of the training set and test on the full validation set.

Figure 6: **Class separation positively correlates with accuracy when relearning how to classify ImageNet classes from limited data.** All accuracy numbers are computed on a 10,000 example subset of the ImageNet validation set (10 examples per class). Blue dots indicate accuracy of the weights of the original ImageNet-trained model. Orange dots indicate accuracy of a linear classifier trained on the other 40,000 examples from the ImageNet validation set. Lines connect blue and orange dots corresponding to the same objective. The gap between the accuracies of the original and relearned classifiers narrows as class separation increases. Class separation is measured on the ImageNet training set. All numbers are averaged over 8 models.

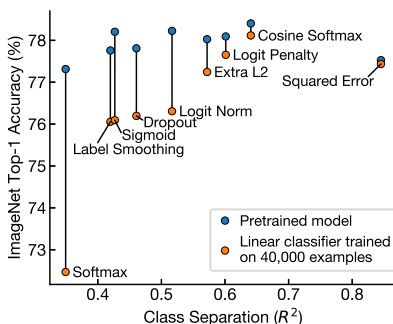

seems that representations with greater class separation are "overfit," not to the pretraining datapoints, but to the pretraining *classes*—they perform better for classifying these classes, but worse when the downstream task requires classifying different classes.

Our results above provide some further evidence that greater class separation can be beneficial in real-world scenarios where downstream datasets share classes with the pretraining dataset. In Tables 1 and 4, representations with the lowest class separation perform best on all datasets except for Oxford-IIIT Pets, where representations with slightly greater class separation consistently perform slightly better. We thus explore class overlap between transfer datasets and ImageNet in Appendix F, and find that, of the 37 cat and dog breeds in Oxford-IIIT Pets, 25 correspond directly to ImageNet classes. Furthermore, it is possible to achieve 71.2% accuracy on Oxford-IIIT Pets simply by taking the top-1 predictions of an ImageNet classifier and and mapping them directly to its classes, but this strategy achieves much lower accuracy on other datasets.

# 4 Related work

**Understanding training objectives.** There is a great deal of previous work that investigates why some objectives perform better than others, using both theoretical and empirical approaches. On linearly separable data, theoretical analysis shows that gradient descent on both unregularized logistic or multinomial logistic regression objectives (i.e., linear models with sigmoid or softmax cross-entropy loss) eventually converges to the minimum norm solution [63]. These results can be extended to neural networks in certain restricted settings [28, 74]. However, the convergence to this solution is very slow. Theoretical analysis of dropout has bounded the excess risk of single-layer [70] and two-layer [42] networks. Empirical studies have attempted to understand regularization in more realistic settings. Label smoothing has been explained in terms of mitigation of label noise [39, 13], entropy regularization [52, 41], and accelerated convergence [78]. Other work has shown that dropout on intermediate layers has both implicit and explicit effects, and both are important to accuracy [74]. Janocha and Czarnecki [32] previously studied both empirical and mathematical similarities and differences among loss functions for supervised classification in the context of deep learning.

**Training objectives for transfer.** Our study of the transferability of networks trained with different objectives extends the previous investigation of Kornblith et al. [34], which showed that two of the regularizers considered here (label smoothing and dropout) lead to features that transfer worse. Similar results have been reported for self-supervised learning, where the loss parameters that maximize accuracy on the contrastive task do not provide the best features for linear classification [14]. Our work demonstrates that this phenomenon is pervasive, and connects it to properties of hidden representations. By contrast, work that explores out-of-distribution (OOD) generalization, where the training and evaluation datasets consist of the same classes but with some degree of distribution shift, finds that ImageNet accuracy is highly predictive of OOD accuracy across many pretrained models [55, 67]. In Appendix Table C.2, we show that, across loss functions, ImageNet validation set accuracy is not entirely correlated with OOD accuracy, but different losses produce much smaller differences in OOD accuracy than linear transfer accuracy.

Other related work has attempted to devise loss functions to learn more transferable embeddings for few-shot learning [62, 65, 7, 15, 49], but embeddings of models trained with softmax cross-entropy often perform on par with these more sophisticated techniques [68]. Doersch et al. [22] motivate their few-shot learning method as a way to mitigate "supervision collapse," which they do not directly quantify, but is closely related to our notion of class separation. While our paper was under review,

two studies reported advantages for increased final layer $L^2$ regularization when performing few-shot linear transfer from very large pretraining datasets to some, but not all, downstream tasks [80, 1]. Our findings suggest that these advantages may arise because the pretraining and downstream tasks share classes, but further investigation is needed to confirm this hypothesis.

**Class separation.** Prior work has investigated class separation in neural networks in a variety of different ways. Theoretical work connects various concepts of the normalized margin between classes to the generalization properties of neural networks [4, 44, 5, 45]. Empirically, Chen et al. [12] measure class separation via "angular visual hardness," the arccosine-transformed cosine similarity between the weight vectors and examples, and suggested an association between this metric and accuracy. However, as shown in Appendix Figure E.1, this metric is unable to differentiate between networks trained with softmax and sigmoid cross-entropy. Papyan et al. [50] describe a phenomenon they call "neural collapse," which occurs after training error vanishes and is associated with the collapse of the class means, classifier, and activations to the vertices of an equiangular tight frame, implying maximal class separation. We relate our observations to theirs in Appendix Table E.2. More generally, studies have investigated how class information evolves through the hidden layers of neural networks using linear classifiers [2], binning estimators of mutual information [61, 58, 26], Euclidean distances [59], the soft nearest neighbor loss [24], and manifold geometry [3, 16]. In the context of a study of representational consistency across networks trained from different initializations, Mehrer et al. [40] previously reported increased class separation in CIFAR-10 networks trained with dropout.

Other work has explored relationships between measures of class separation in embedding spaces and accuracy in few-shot and deep metric learning settings. In deep metric learning, Roth et al. [56] show that objectives that lead to greater class separation generally produce higher recall, and find a similar relationship for a measure of the uniformity of the singular value spectrum of the representation space. In few-shot learning, different work has reached different conclusions: Goldblum et al. [25] find that regularization that *increases* class separation improves few-shot classification performance on mini-ImageNet and CIFAR-FS, whereas Liu et al. [37] find that introducing a negative margin into a cosine softmax loss yields both lower class separation and higher accuracy.

## 5   Limitations

Although we investigate many losses and multiple architectures, our experiments are limited to moderately sized datasets with moderately sized models, and our conclusions are limited to supervised classification settings. ResNet-50 on trained on ImageNet is a realistic transfer scenario where our analysis is computationally tractable, but bigger models trained on bigger datasets with more classes achieve better performance. These bigger datasets with richer label spaces could potentially help to mitigate the trade-off between pretraining accuracy and feature quality, and may also have class-level overlap with many downstream tasks, making greater class separation helpful rather than harmful.

Our results establish that a wide variety of losses that improve over vanilla softmax cross-entropy lead to greater class separation, but this observation does not immediately lead to a recipe for higher pretraining accuracy. The relationship between class separation and pretraining accuracy is non-monotonic: squared error achieves the highest class separation but does not significantly outperform vanilla softmax cross-entropy.

## 6   Conclusion

In this study, we find that the properties of a loss that yields good performance on the pretraining task are different from the properties of a loss that learns good generic features. Training objectives that lead to better performance on the pretraining task learn more invariant representations with greater class separation. However, these same properties are detrimental when fixed features are transferred to other tasks. Moreover, because different loss functions produce different representations only in later network layers, which change substantially during fine-tuning, gains in pretraining accuracy do not lead to gains when models are fine-tuned on downstream tasks.

Our work suggests opportunities for improving fixed feature representations in deep neural networks. We see no inherent reason that features learned by softmax cross-entropy should be optimal for transfer, but previous work has optimized for pretraining accuracy, rather than transferable features. With the increasing importance of transfer learning in deep learning, we believe that future research into loss functions should explicitly target and evaluate performance under transfer settings.

## Acknowledgements

We thank Pieter-Jan Kindermans for comments on the manuscript, and Geoffrey Hinton and David Fleet for useful discussions.

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
