# Appendix

# A  Details of training and hyperparameter tuning

## A.1  Training and tuning neural networks

We trained ImageNet models (ResNet-50 [9, 7, 6] "v1.5"[3]) with SGD with Nesterov momentum of 0.9 and a batch size 4096 and weight decay of $8 \times 10^{-5}$ (applied to the weights but not batch norm parameters). After 10 epochs of linear warmup to a maximum learning rate of 1.6, we decayed the learning rate by a factor of 0.975 per epoch. We took an exponential moving average of the weights over training as in Szegedy et al. [30], with a momentum factor of 0.9999. We used standard data augmentation comprising random crops of 10-100% of the image with aspect ratios of 0.75 to 1.33 and random horizontal flips. At test time, we resized images to 256 pixels on their shortest side and took a $224 \times 224$ center crop. Each training run took approximately 1.5 hours on a 128-core TPU v2 node. Overall, the experiments in the main text reflect 72 total training runs, plus an approximately equal number of training runs used to tune hyperparameters.

To tune hyperparameters, we held out a validation set of 50,046 ImageNet training examples. We initially performed a set of training runs with a wide range of different parameters, and then narrowed the hyperparameter range to the range shown in Table A.1. To further tune the hyperparameters and the epoch for early stopping, we performed 3 training runs per configuration.[4] After determining the hyperparameters, we trained models on the full training set. We note that early stopping is important to achieve maximal performance with our learning rate schedule, but does not affect the conclusions we draw regarding transferability and class separation, as we confirm in Appendices D.2 and E.4.

Table A.1: **Hyperparameters for ResNet-50 on ImageNet.**

| Loss/regularizer | Hyperparameters | Epochs |
|---|---|---|
| Softmax | N/A | 146 |
| Label smoothing | $\alpha = \{0.08, 0.09, \mathbf{0.1}, 0.11.0.12\}$ | 180 |
| Sigmoid | N/A | 166 |
| Extra final layer $L^2$ | $\lambda_{\text{final}} = \{4e\text{-}4, 6e\text{-}4, \mathbf{8e\text{-}4}, 1e\text{-}3\}$ | 168 |
| Dropout | $\rho = \{0.6, 0.65, \mathbf{0.7}, 0.75, 0.8, 0.85\}$ | 172 |
| Logit penalty | $\beta = \{5e\text{-}5, 1e\text{-}4, 2e\text{-}4, 4e\text{-}4, \mathbf{6e\text{-}4}, 8e\text{-}4\}$ | 180 |
| Logit normalization | $\tau = \{0.03, \mathbf{0.04}, 0.05, 0.06\}$ | 152 |
| Cosine softmax | $\tau = \{0.04, 0.045, \mathbf{0.05}, 0.06, 0.07, 0.08\}$ | 158 |
| Squared error | $\kappa = 9$, $M = 60$, loss scale $= 10$ | 196 |

## A.2  Training and tuning multinomial logistic regression classifiers

To train multinomial logistic regression classifiers on fixed features, we use L-BFGS [24], following a similar approach to previous work [16, 27]. We first extracted features for every image in the training set, by resizing them to 224 pixels on the shortest side and taking a $224 \times 224$ pixel center crop. We held out a validation set from the training set, and used this validation set to select the $L^2$ regularization hyperparameter, which we selected from 45 logarithmically spaced values between $10^{-6}$ and $10^5$, applied to the sum of the per-example losses. Because the optimization problem is convex, we used the previous weights as a warm start as we increased the $L^2$ regularization hyperparameter. After finding the optimal hyperparameter on this validation set, we retrained on the training + validation sets and evaluated accuracy on the test set. We measured either top-1 or mean per-class accuracy, depending on which was suggested by the dataset creators. See Table A.2 for further details of the datasets investigated.

---

[3]The `torchvision` ResNet-50 model and the "official" TensorFlow ResNet both implement this architecture, which was first proposed by Gross and Wilber [7] and differs from the ResNet v1 described by He et al. [9] in performing strided convolution in the first $3 \times 3$ convolution in each stage rather than the first $1 \times 1$ convolution. Our implementation initializes the $\gamma$ parameters of the last batch normalization layer in each block to 0, as in Goyal et al. [6].

[4]Due to the large number of hyperparameter configurations, for squared error, we performed only 1 run per configuration to select hyperparameters, but 3 to select the epoch at which to stop. We manually narrowed the hyperparameter search range until all trained networks achieved similar accuracy. The resulting hyperparaameters performed better than those suggested by Hui and Belkin [13].

Table A.2: **Datasets examined in transfer learning.**

| Dataset | Classes | Size (train/test) | Accuracy measure |
|---|---|---|---|
| Food-101 [2] | 101 | 75,750/25,250 | top-1 |
| CIFAR-10 [18] | 10 | 50,000/10,000 | top-1 |
| CIFAR-100 [18] | 10 | 50,000/10,000 | top-1 |
| Birdsnap [1] | 500 | 47,386/2,443 | top-1 |
| SUN397 [32] | 397 | 19,850/19,850 | top-1 |
| Stanford Cars [17] | 196 | 8,144/8,041 | top-1 |
| Oxford-IIIT Pets [26] | 37 | 3,680/3,369 | mean per-class |
| Oxford 102 Flowers [23] | 102 | 2,040/6,149 | mean per-class |

## A.3 Fine-tuning

In our fine-tuning experiments in Table 2, we used standard ImageNet-style data augmentation and trained for 20,000 steps with SGD with momentum of 0.9 and cosine annealing [20] without restarts. We performed hyperparameter tuning on a validation set, selecting learning rate values from a logarithmically spaced grid of 8 values between $10^{-5.5}$ and $10^{-1}$ and weight decay values from a logarithmically spaced grid of 8 values between $10^{-6.5}$ and $10^{-3}$, as well as no weight decay, dividing the weight decay by the learning rate. We manually verified that optimal hyperparameter combinations for each loss and dataset fall inside this grid. We averaged the accuracies obtained by hyperparameter tuning over 3 runs starting from 3 different pretrained ImageNet models and picked the best. We then retrained each model on combined training + validation sets and tested on the provided test sets.

## B   Confirmation of main findings with Inception v3

To confirm that our findings hold across architectures, we performed experiments using Inception v3 [30], which does not have residual connections but still attains good performance on ImageNet ILSVRC. Because our goal was to validate the consistency of our observations, rather than to achieve maximum accuracy, we used the same hyperparameters as for ResNet-50, but selected the epoch for early stopping on a holdout set.

Table B.1 confirms our main findings involving class separation and transfer accuracy. As in Table 1, we observe that softmax learns more transferable features than other loss functions, and as in Table 3, we find that lower class separation is associated with greater transferability. Figure B.1 confirms our finding that the choice of loss function affects representations only in later layers of the network.

Table B.1: **Objectives that produce higher ImageNet accuracy lead to less transferable fixed features, for Inception v3.** "ImageNet" columns reflect accuracy of each model on the ImageNet validation set. "Transfer" columns reflect accuracy of $L^2$-regularized multinomial logistic regression classifiers trained to classify different transfer datasets using the fixed penultimate layer features of the ImageNet-trained networks. Numbers are averaged over 3 different pretraining initializations, and all values not significantly different than the best are bold-faced ($p < 0.05$, $t$-test). The strength of $L^2$ regularization is selected on a validation set that is independent of the test set. See Table 1 for results with ResNet-50.

| | ImageNet | | | Transfer | | | | | | | |
|---|---|---|---|---|---|---|---|---|---|---|---|
| Loss | Top-1 | Top-5 | $R^2$ | Food | CIFAR10 | CIFAR100 | Birdsnap | SUN397 | Cars | Pets | Flowers |
| Softmax | 78.6 | 94.24 | 0.356 | **74.5** | **92.4** | **76.2** | **59.3** | **63.1** | **64.4** | 92.2 | **94.0** |
| Label smoothing | 78.8 | **94.60** | 0.441 | 73.3 | 91.6 | **75.0** | 56.1 | 62.4 | 60.3 | **93.0** | 92.4 |
| Sigmoid | **79.1** | 94.17 | 0.444 | 73.7 | 91.3 | 74.7 | 55.0 | 62.0 | 60.7 | **92.8** | **93.0** |
| More final layer $L^2$ | **79.0** | 94.52 | 0.586 | 70.1 | 91.0 | 73.3 | 52.4 | 61.0 | 51.1 | **92.5** | 89.6 |
| Dropout | **79.0** | 94.50 | 0.454 | 72.6 | 91.5 | 74.7 | 56.3 | 62.1 | 59.2 | **92.7** | 92.2 |
| Logit penalty | 78.9 | **94.63** | 0.638 | 69.1 | 90.6 | 72.1 | 49.3 | 59.2 | 52.3 | 92.3 | 87.9 |
| Logit normalization | 78.8 | 94.34 | 0.559 | 67.4 | 90.6 | 72.2 | 50.9 | 58.5 | 45.6 | 92.1 | 84.2 |
| Cosine softmax | 78.9 | 94.38 | 0.666 | 63.1 | 90.3 | 71.5 | 45.8 | 55.6 | 38.0 | 90.6 | 75.2 |
| Squared error | 77.7 | 93.28 | 0.838 | 45.3 | 84.1 | 57.6 | 25.0 | 41.1 | 18.8 | 85.7 | 54.8 |

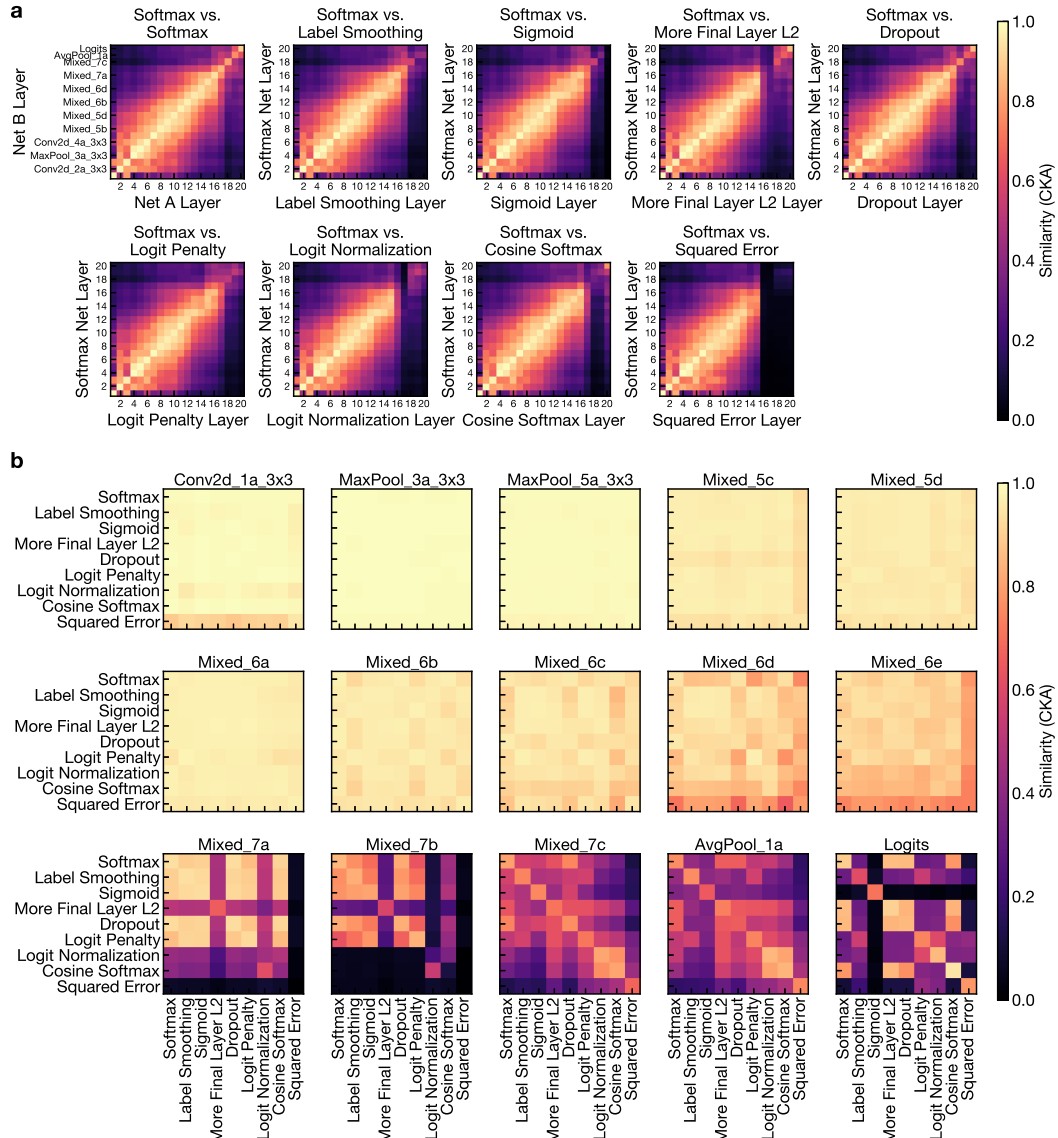

Figure B.1: **The choice of loss function affects representations only in later network layers, for Inception v3.** All plots show linear centered kernel alignment (CKA) between representations computed on the ImageNet validation set. **a**: CKA between network layers, for pairs of Inception v3 models trained from different initializations with the same or different losses. **b**: CKA between representations extracted from corresponding layers of networks trained with different loss functions. Diagonal reflects similarity of networks with the same loss function trained from different initalizations. See Figure 2 for results with ResNet-50.

# C   Additional evaluation of regularizers and losses

## C.1   Training accuracy and learning curves

Table C.1: **Training accuracy of ResNet-50 models.** Regularizers and modified losses resulted in lower ImageNet training set accuracy, consistent with the notion that regularization sacrifices training accuracy to attain greater test accuracy. However, label smoothing was statstically tied with vanilla softmax cross-entropy in terms of training top-1 accuracy, and performed slightly better in terms of training top-5 accuracy. See Table 1 for validation set accuracy.

| Loss/regularizer | Top-1 Acc. (%) | Top-5 Acc. (%) |
|---|---|---|
| Softmax | $93.61 \pm 0.01$ | $99.33 \pm 0.002$ |
| Label smoothing | $93.62 \pm 0.04$ | $99.43 \pm 0.007$ |
| Sigmoid | $93.22 \pm 0.01$ | $99.19 \pm 0.002$ |
| Extra final layer $L^2$ | $91.62 \pm 0.01$ | $98.85 \pm 0.003$ |
| Dropout | $92.25 \pm 0.01$ | $99.03 \pm 0.003$ |
| Logit penalty | $93.04 \pm 0.01$ | $99.13 \pm 0.002$ |
| Logit normalization | $92.86 \pm 0.01$ | $99.01 \pm 0.003$ |
| Cosine softmax | $92.47 \pm 0.01$ | $98.75 \pm 0.004$ |
| Squared error | $91.65 \pm 0.01$ | $98.59 \pm 0.002$ |

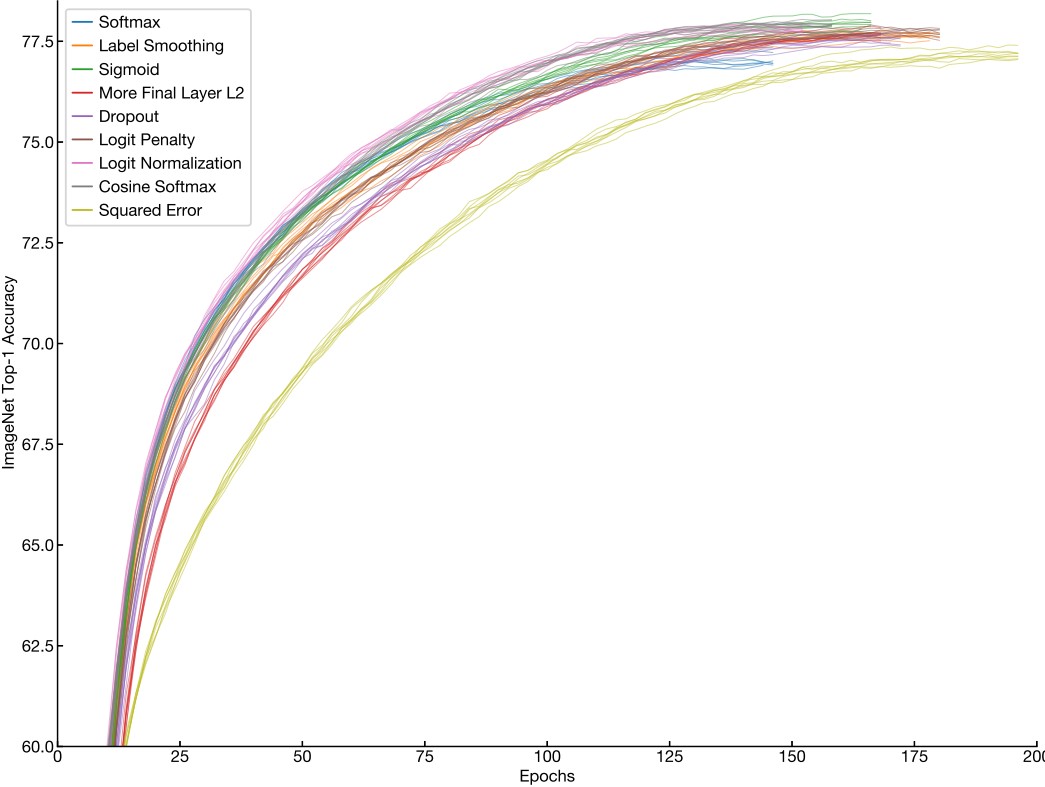

Figure C.1: **Evolution of ImageNet validation accuracy over training.** Each curve represents a different model. For each loss function, curves terminate at the epoch that provided the highest holdout set accuracy. Validation accuracy rises rapidly due to the use of an exponential moving average of the weights for evaluation. Some loss functions, such as logit normalization, appear to provide higher accuracy than vanilla softmax cross-entropy over the entire training run.

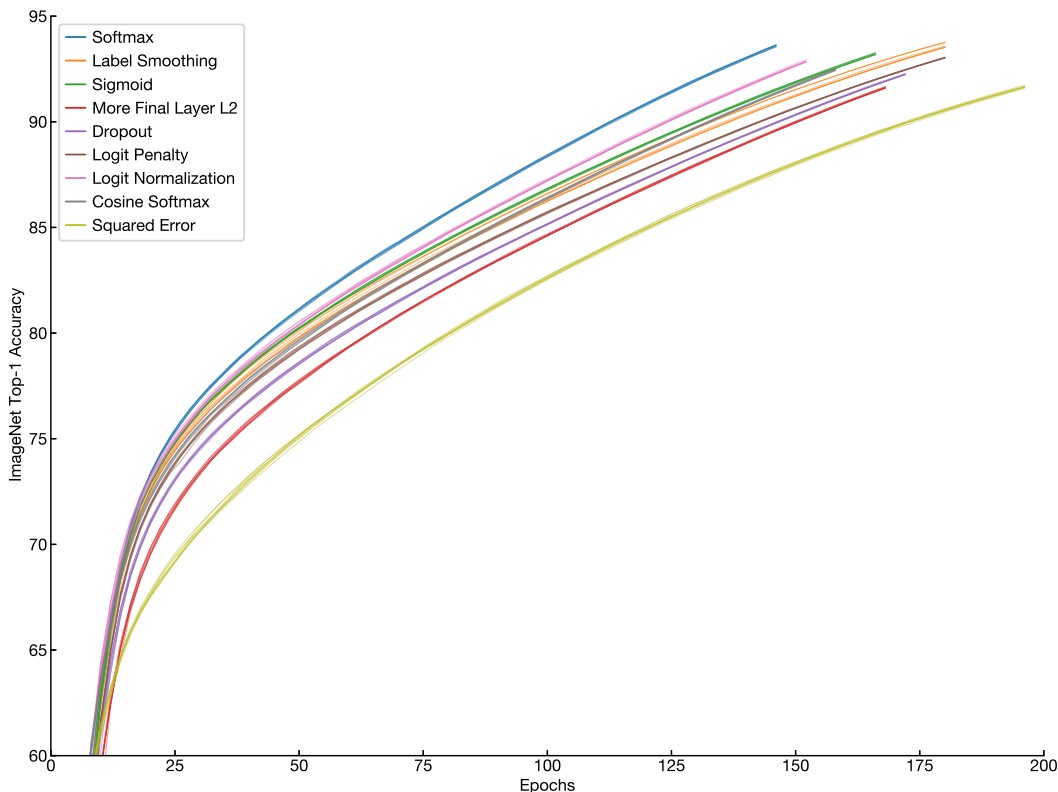

Figure C.2: **Evolution of ImageNet training accuracy.** Each curve represents a different model. For each loss function, curves terminate at the epoch that provided the highest holdout set accuracy.

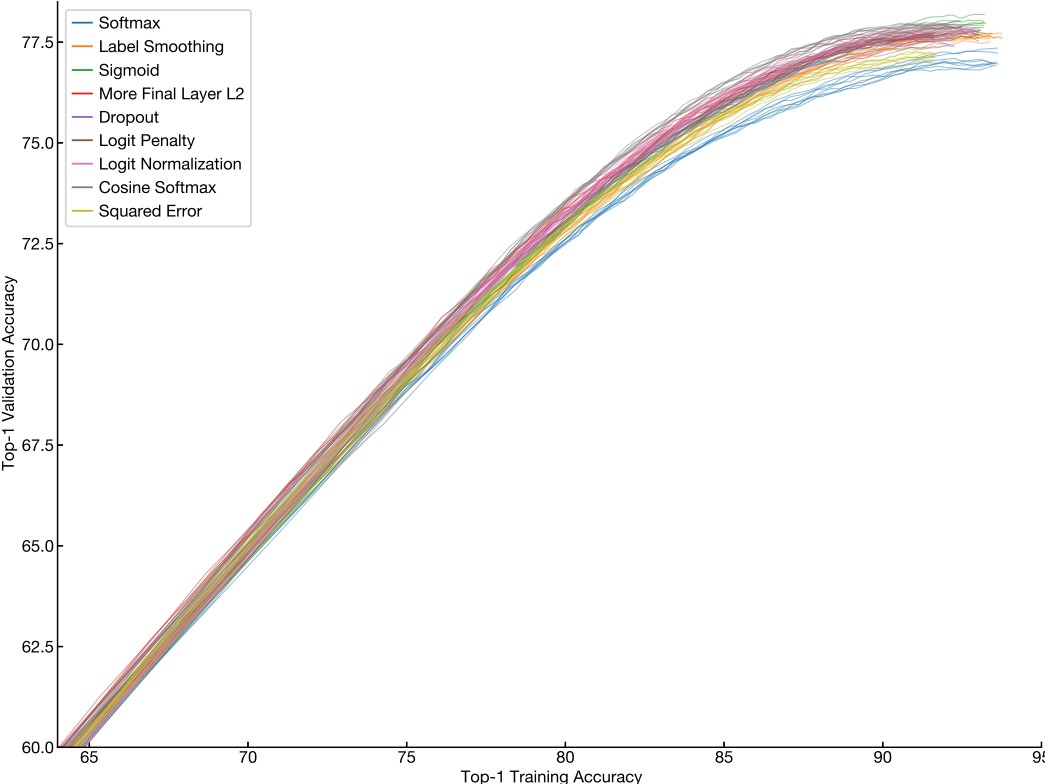

Figure C.3: **Validation versus training accuracy.** Each curve represents a different model. For each loss function, curves terminate at the training accuracy that provided the highest holdout set accuracy. Regularized models achieve higher validation accuracy at a given training accuracy as compared to softmax.

## C.2 Robustness and calibration

In addition to the differences in class separation and accuracy described in the text, losses differed in out-of-distribution robustness, and in the calibration of the resulting predictions. Table C.2 shows results for ImageNet-trained ResNet-50 models on the out-of-distribution test sets ImageNet-V2 [29], ImageNet-A [11], ImageNet-Sketch [31], ImageNet-R [12], and ImageNet-C [10]. In almost all cases, alternative loss functions outperformed softmax cross-entropy, with logit normalization and cosine softmax typically performing slightly better than alternatives. Effects on calibration, shown in Table C.3, were mixed. Label smoothing substantially reduced expected calibration error [8], as previously shown by Müller et al. [22], although cosine softmax achieved a lower negative log likelihood. However, there was no clear relationship between calibration and accuracy. Logit penalty performed well in terms of accuracy, but provided the worst calibration of any objective investigated.

Table C.2: **Regularizers and alternative losses improve performance on out-of-distribution test sets.** Accuracy averaged over 8 ResNet-50 models per loss.

| Loss/regularizer | ImageNet-V2 (%) | ImageNet-A (%) | IN-Sketch (%) | ImageNet-R (%) | ImageNet-C (mCE) |
|---|---|---|---|---|---|
| Softmax | $65.0 \pm 0.1$ | $2.7 \pm 0.0$ | $21.8 \pm 0.1$ | $36.8 \pm 0.1$ | $75.9 \pm 0.1$ |
| Label smoothing | $\mathbf{65.7} \pm 0.1$ | $3.8 \pm 0.1$ | $22.5 \pm 0.1$ | $37.8 \pm 0.1$ | $75.2 \pm 0.1$ |
| Sigmoid | $\mathbf{65.9} \pm 0.1$ | $3.3 \pm 0.0$ | $22.6 \pm 0.1$ | $36.6 \pm 0.1$ | $74.6 \pm 0.1$ |
| Extra final layer $L^2$ | $\mathbf{65.8} \pm 0.1$ | $3.3 \pm 0.0$ | $23.1 \pm 0.1$ | $37.7 \pm 0.1$ | $74.1 \pm 0.1$ |
| Dropout | $65.4 \pm 0.0$ | $3.1 \pm 0.1$ | $23.0 \pm 0.1$ | $37.2 \pm 0.1$ | $74.5 \pm 0.1$ |
| Logit penalty | $\mathbf{65.8} \pm 0.0$ | $4.5 \pm 0.0$ | $22.8 \pm 0.1$ | $38.1 \pm 0.1$ | $74.3 \pm 0.1$ |
| Logit normalization | $\mathbf{65.8} \pm 0.1$ | $\mathbf{4.8} \pm 0.1$ | $23.7 \pm 0.1$ | $\mathbf{39.2} \pm 0.1$ | $73.2 \pm 0.1$ |
| Cosine softmax | $\mathbf{65.8} \pm 0.1$ | $4.6 \pm 0.1$ | $\mathbf{24.8} \pm 0.1$ | $38.7 \pm 0.1$ | $\mathbf{72.5} \pm 0.1$ |
| Squared error | $65.3 \pm 0.1$ | $4.5 \pm 0.1$ | $22.4 \pm 0.1$ | $36.3 \pm 0.1$ | $74.6 \pm 0.1$ |

Table C.3: **Some regularizers and alternative losses improve calibration.** We report negative log likelihood (NLL) and expected calibration error (ECE), averaged over 3 ResNet-50 models trained with each loss on the ImageNet validation set, before and after scaling the temperature of the probability of the distribution to minimize NLL, as in Guo et al. [8]. These models were trained with a holdout set of 50,046 ImageNet training examples, which were then used to perform temperature scaling to minimize NLL. ECE is computed with 15 evenly spaced bins. For networks trained with sigmoid loss, we normalize the probability distribution by summing probabilities over all classes.

| Loss/regularizer | Uncalibrated | | With temperature scaling | |
|---|---|---|---|---|
| | NLL | ECE | NLL | ECE |
| Softmax | $0.981 \pm 0.002$ | $0.073 \pm 0.0001$ | $0.917 \pm 0.002$ | $0.027 \pm 0.0004$ |
| Label smoothing | $0.947 \pm 0.001$ | $\mathbf{0.016} \pm 0.0007$ | $0.941 \pm 0.001$ | $0.044 \pm 0.0004$ |
| Sigmoid | $0.944 \pm 0.002$ | $0.044 \pm 0.0003$ | $0.914 \pm 0.002$ | $\mathbf{0.019} \pm 0.0002$ |
| Extra final layer $L^2$ | $0.976 \pm 0.002$ | $0.081 \pm 0.0003$ | $0.908 \pm 0.002$ | $0.038 \pm 0.0006$ |
| Dropout | $0.971 \pm 0.002$ | $0.074 \pm 0.0009$ | $0.905 \pm 0.002$ | $0.031 \pm 0.0002$ |
| Logit penalty | $1.041 \pm 0.001$ | $0.090 \pm 0.0003$ | $0.995 \pm 0.001$ | $0.055 \pm 0.0004$ |
| Logit normalization | $0.965 \pm 0.001$ | $0.069 \pm 0.0002$ | $0.949 \pm 0.001$ | $0.049 \pm 0.0003$ |
| Cosine softmax | $\mathbf{0.912} \pm 0.002$ | $0.066 \pm 0.0006$ | $\mathbf{0.895} \pm 0.002$ | $0.043 \pm 0.0008$ |

## C.3 Similarity of model predictions

Given that many loss functions resulted in similar improvements in accuracy over softmax loss, we sought to determine whether they also produced similar effects on network predictions. For each pair of models, we selected validation set examples that both models classified incorrectly, and measured the percentage of these examples for which the models gave the same prediction. As shown in Figure C.4, models' predictions cluster into distinct groups according to their objectives. Models trained with the same objective (from different initializations) are more similar than models trained with different objectives. In addition, models trained with (regularized) softmax loss or sigmoid loss are more similar to each other than to models trained with logit normalization or cosine softmax, and networks trained with squared error are dissimilar to all others examined. Figure C.5 shows that other ways of measuring the similarity of models' predictions yielded qualitatively similar results.

Although it was possible to identify the loss used to train individual models from their predictions, models trained with the same loss nonetheless disagreed on many examples. Standard deviations in

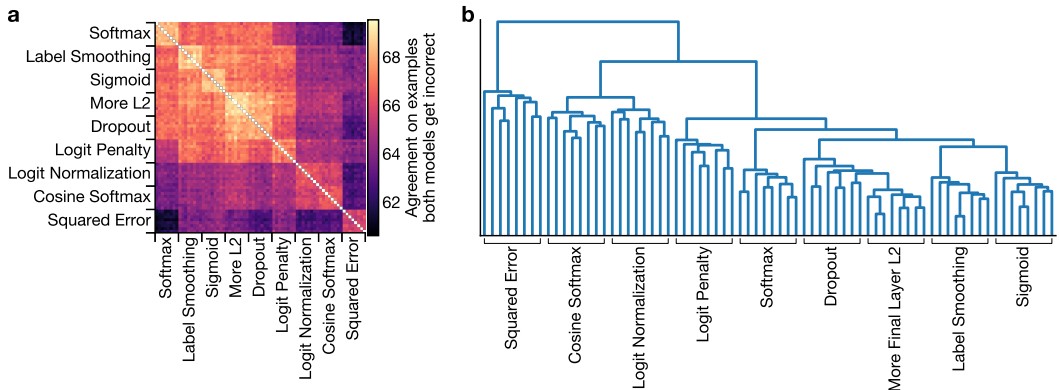

Figure C.4: **Different losses produce different predictions, even when accuracies are close. a**: For each pair of models, we take examples incorrectly classified by both and measure the percentage where the models' top-1 predictions agree. We show results for 8 different initializations trained with each objective. See Figure C.5 for qualitatively similar plots that show percentages of all examples on which models agree, and percentages of images where both models are either correct or incorrect. **b**: Dendrogram based on similarity matrix. All models naturally cluster according to loss, except for "Dropout" and "More Final Layer L2" models.

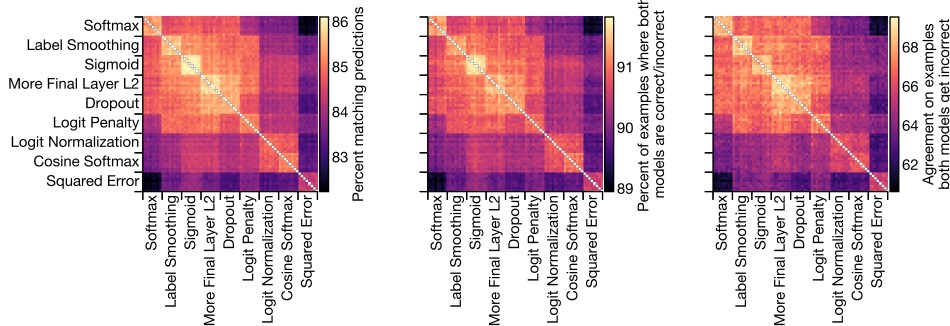

Figure C.5: **Different ways of measuring similarity of single-model ResNet-50 predictions yield similar qualitative results.** In the left panel, we compute the top-1 predictions for pairs of models on the ImageNet validation set and determine the percentage of examples where these predictions match. In the middle panel, we measure the percentage of examples where models either get both right or both wrong. In the right panel, we restrict our analysis to examples that both models get incorrect, and measure the percentage of these examples where both models make the same (incorrect) top-1 prediction.

top-1 accuracy are <0.2% for all losses, but even the most similar pair of models provides different predictions on 13.9% of all validation set examples (Figure C.5). Ensembling can substantially reduce the level of disagreement between models and objectives: When ensembling the 8 models trained with the same loss, the least similar losses (softmax and squared error) disagree on only 11.5% of examples (Figure C.6). However, there was little accuracy benefit to ensembling models trained with different objectives over ensembling models trained with the same objective (Figure C.7).

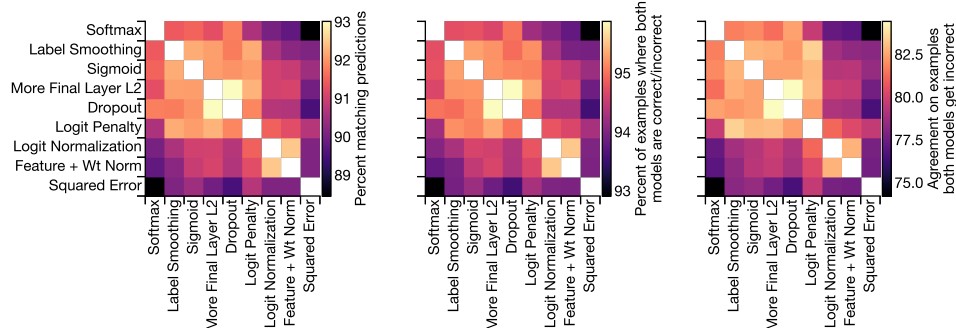

Figure C.6: **Ensemble predictions are substantially more similar than single-model predictions.** Predictions of the ensemble were computed by taking 8 ResNet-50 models trained from different random initializations with the same loss and picking the most common top-1 prediction for each example.

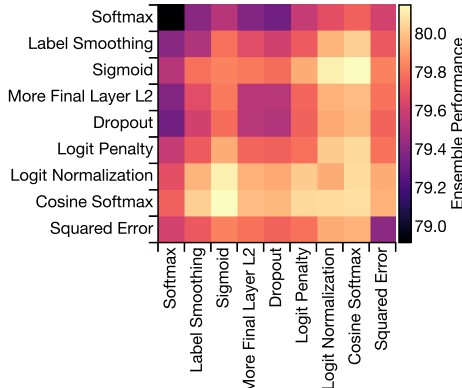

Figure C.7: **Ensembling models trained with different losses provides only modest performance benefits.** Ensembles consist of 8 ResNet-50 models, half of which are trained with the objective on the x-axis, the other half with the objective on the y-axis. The ensemble prediction is the modal class prediction of the 8 models.

## C.4   Combining regularizers does not improve accuracy

Given the clear differences in the effects of different objectives on network predictions, we next asked whether combining regularization or normalization strategies might result in better predictions. Table C.4 shows that these combinations do not improve accuracy. However, as shown in Table C.5, improved data augmentation [4, 33] provides a similar additive gain in accuracy to networks trained with alternative losses as it does to networks trained with softmax cross-entropy. These results suggest that the objectives that improve over softmax cross-entropy do so via similar mechanisms, but data augmentation acts differently.

With longer training, both sigmoid cross-entropy and cosine softmax achieve state-of-the-art accuracy among ResNet-50 networks trained with AutoAugment (Table C.6), matching or outperforming supervised contrastive learning [15]. Combining cosine softmax loss, AutoAugment, and Mixup, we achieve 79.1% top-1 accuracy and 94.5% top-5 accuracy, which was, at the time this paper was first posted, the best reported $224 \times 224$ pixel single-crop accuracy with an unmodified ResNet-50 architecture trained from scratch.

Table C.4: **Combining final-layer regularizers and/or improved losses does not meaningfully enhance performance.** Accuracy of ResNet-50 models on our ImageNet holdout set when combining losses and regularizers between models. All results reflect the maximum accuracy on the holdout set at any point during training, averaged across 3 training runs. Accuracy numbers are higher on the holdout set than the official ImageNet validation set. This difference in accuracy is likely due to a difference in image distributions between the ImageNet training and validation sets, as previously noted in Section C.3.1 of Recht et al. [29].

| | Baseline | Label smoothing $(\alpha = 0.1)$ | Sigmoid | Cosine softmax $(\tau = 0.05)$ |
|---|---|---|---|---|
| Baseline | 79.9 | 80.4 | 80.6 | 80.6 |
| Dropout $(\beta = 0.7)$ | 80.3 | 80.3 | 80.3 | 80.2 |
| Dropout $(\beta = 0.8)$ | 80.2 | 80.4 | 80.4 | 80.4 |
| Dropout $(\beta = 0.9)$ | | 80.3 | 80.5 | 80.6 |
| Dropout $(\beta = 0.95)$ | | 80.4 | 80.6 | 80.7 |
| Logit penalty $(\gamma = 5 \times 10^{-5})$ | 80.4 | 80.3 | 80.5 | 80.6 |
| Logit penalty $(\gamma = 1 \times 10^{-4})$ | 80.4 | 80.3 | 80.5 | 80.5 |
| Logit penalty $(\gamma = 2 \times 10^{-4})$ | 80.4 | 80.3 | 80.4 | 80.5 |
| Logit penalty $(\gamma = 4 \times 10^{-4})$ | 80.4 | 80.2 | 80.3 | 80.5 |
| Logit penalty $(\gamma = 6 \times 10^{-4})$ | 80.5 | 80.2 | 80.3 | 80.5 |
| Logit normalization $(\tau = 0.02)$ | | 80.0 | 80.4 | |
| Logit normalization $(\tau = 0.03)$ | 80.3 | 80.4 | 80.6 | |
| Logit normalization $(\tau = 0.04)$ | 80.4 | 80.5 | 80.6 | |
| Logit normalization $(\tau = 0.05)$ | 80.3 | 80.5 | 80.5 | |
| Logit normalization $(\tau = 0.06)$ | 80.3 | 80.4 | 80.5 | |
| Cosine normalization $(\tau = 0.045)$ | 80.6 | | 80.5 | |
| Cosine normalization $(\tau = 0.05)$ | 80.6 | | 80.6 | |
| Cosine normalization $(\tau = 0.06)$ | 80.4 | | 75.3 | |

Table C.5: **AutoAugment and Mixup provide consistent accuracy gains beyond well-tuned losses and regularizers.** Top-1 accuracy of ResNet-50 models trained with and without AutoAugment, averaged over 3 (with AutoAugment) or 8 (without AutoAugment) runs. Models trained with AutoAugment use the loss hyperparameters chosen for models trained without AutoAugment, but the point at which to stop training was chosen independently on our holdout set. For models trained with Mixup, the mixing parameter $\alpha$ is chosen from $[0.1, 0.2, 0.3, 0.4]$ on the holdout set. Best results in each column, as well as results insignificantly different from the best ($p > 0.05$, t-test), are bold-faced.

| | Standard augmentation | | AutoAugment | | Mixup | |
|---|---|---|---|---|---|---|
| Loss/regularizer | Top-1 (%) | Top-5 (%) | Top-1 (%) | Top-5 (%) | Top-1 (%) | Top-5 (%) |
| Softmax | $77.0 \pm 0.06$ | $93.40 \pm 0.02$ | $77.7 \pm 0.05$ | $93.74 \pm 0.05$ | $78.0 \pm 0.05$ | $93.98 \pm 0.03$ |
| Sigmoid | $\mathbf{77.9} \pm 0.05$ | $93.50 \pm 0.02$ | $\mathbf{78.5} \pm 0.04$ | $93.82 \pm 0.02$ | $\mathbf{78.5} \pm 0.07$ | $93.94 \pm 0.04$ |
| Logit penalty | $77.7 \pm 0.02$ | $\mathbf{93.83} \pm 0.02$ | $\mathbf{78.3} \pm 0.05$ | $\mathbf{94.10} \pm 0.03$ | $78.0 \pm 0.05$ | $93.95 \pm 0.05$ |
| Cosine softmax | $\mathbf{77.9} \pm 0.02$ | $\mathbf{93.86} \pm 0.01$ | $\mathbf{78.3} \pm 0.02$ | $\mathbf{94.12} \pm 0.04$ | $\mathbf{78.4} \pm 0.04$ | $\mathbf{94.14} \pm 0.02$ |

Table C.6: **Comparison with state-of-the-art.** All results are for ResNet-50 models trained with AutoAugment. Loss hyperparameters are the same as in Table C.5, but the learning schedule decays exponentially at a rate of 0.985 per epoch, rather than 0.975 per epoch. This learning rate schedule takes approximately $2\times$ as many epochs before it reaches peak accuracy, and provides a $\sim 0.4\%$ improvement in top-1 accuracy across settings.

| Loss | Epochs | Top-1 (%) | Top-5 (%) |
|---|---|---|---|
| Softmax [4] | 270 | 77.6 | 93.8 |
| Supervised contrastive [15] | 700 | **78.8** | 93.9 |
| *Ours:* | | | |
| Softmax | 306 | $77.9 \pm 0.02$ | $93.77 \pm 0.03$ |
| Sigmoid | 324 | $\mathbf{78.9} \pm 0.04$ | $93.96 \pm 0.06$ |
| Logit penalty | 346 | $\mathbf{78.6} \pm 0.07$ | $\mathbf{94.30} \pm 0.01$ |
| Cosine softmax | 308 | $\mathbf{78.7} \pm 0.04$ | $\mathbf{94.24} \pm 0.02$ |
| *Ours (with Mixup):* | | | |
| Sigmoid | 384 | $79.1 \pm 0.06$ | $94.28 \pm 0.03$ |
| Cosine softmax | 348 | $79.1 \pm 0.09$ | $94.49 \pm 0.01$ |

# D Additional transfer learning results

## D.1 Scatterplots of ImageNet vs. transfer accuracy by dataset

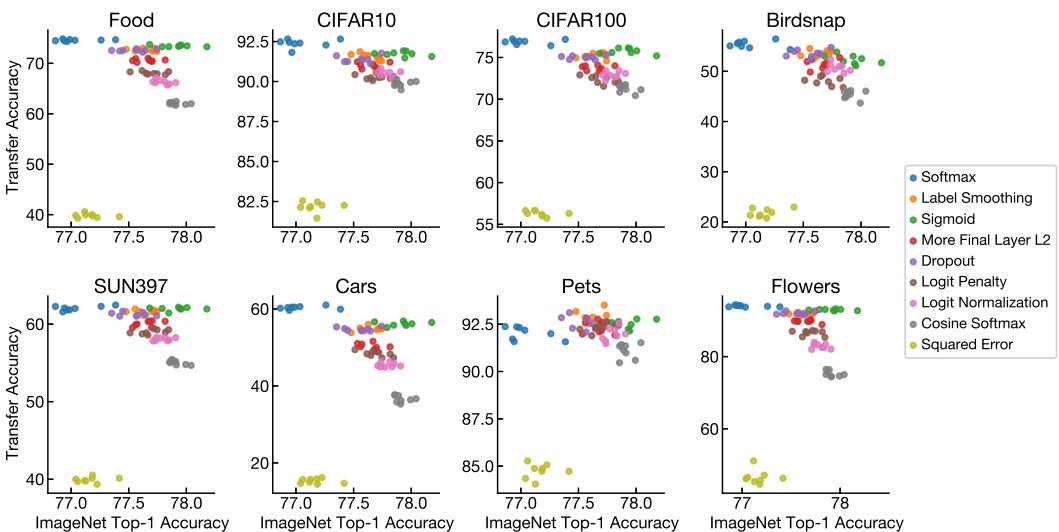

Figure D.1: **Higher ImageNet accuracy is not associated with higher linear transfer accuracy.** Points represent the accuracies of individual training runs. Panels represent different datasets. See Figure 1 for a similar plot of transfer accuracy averaged across datasets.

## D.2 Training dynamics of transfer accuracy

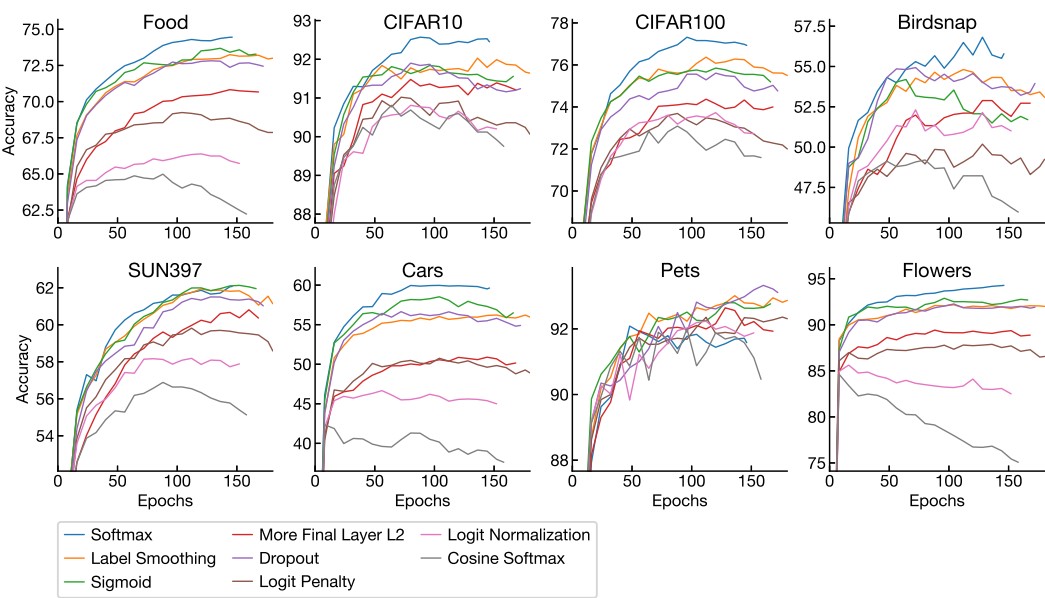

Figure D.2: **On most datasets, softmax cross-entropy achieves the highest linear transfer accuracy over the entire training run.** For each loss function, we evaluate the linear transfer accuracy of a ResNet-50 model every 6 epochs over the course of a single ImageNet pretraining run. Lines terminate at the final checkpoint, selected as described in Appendix A.1. Conclusions regarding the superiority of different loss functions match those from Table 1: Vanilla softmax cross-entropy achieves greater accuracy than other losses except on SUN397, where it is tied with sigmoid cross-entropy, and Pets, where other losses perform better. We do not show results for squared error because they are off the scale of the plots for all datasets except Pets.

## D.3 Results on Chexpert

In Table D.1, we evaluate the performance of linear classifiers trained to classify the Chexpert chest X-ray dataset [14] based on the penultimate layer representations of our ImageNet-pretrained models, using the procedure described in Appendix A.2. We treat both uncertain and unmentioned observations as negative. We tune the $L^2$ regularization hyperparameter separately for each class. We approximate AUC using 1000 evenly-spaced bins. The official validation set of 234 images is very small and results in high variance; vanilla softmax cross-entropy achieves the best numerical results on all but one pathology, but many losses are statistically tied. We thus examine a second setting where we split 22,431 images from the training set and evaluate on these images. On this split, we find that softmax cross-entropy performs significantly better than all other losses on 4 of the 5 pathologies, and is tied for the best AUC on the fifth.

We note that the domain shift between Chexpert and ImageNet is very large. Given the extent of the domain shift, linear transfer will always perform far worse than fine-tuning. However, fine-tuning is unlikely to reveal differences among losses, particularly given that Raghu et al. [28] previously reported that ImageNet pretraining provides no accuracy advantage over training from scratch on this dataset. Nonetheless, we find that, even in this somewhat extreme setting, the fixed features learned by vanilla softmax cross-entropy on ImageNet work better than features learned by other losses.

Table D.1: **Transfer learning results on Chexpert.** AUC of classifiers learned using $L^2$-regularized multinomial logistic regression on the fixed penultimate layer features of the ImageNet-trained networks. Numbers are averaged over 8 different pretraining initializations, and all values not significantly different than the best are bold-faced ($p < 0.05$, $t$-test). The strength of $L^2$ regularization is selected on a validation set that is independent of the test set. See Table 1 for results for natural image datasets.

| Pretraining loss | Atelectasis | Cardiomegaly | Consolidation | Edema | Pleural effusion |
|---|---|---|---|---|---|
| *Official validation set (234 images):* | | | | | |
| Softmax | **74.9** | **75.3** | **86.9** | **87.4** | **86.4** |
| Label smoothing | **74.0** | **73.6** | **85.8** | **86.5** | **85.8** |
| Sigmoid | **74.6** | **75.0** | **86.9** | **87.3** | 85.3 |
| More final layer $L^2$ | **74.6** | **74.1** | **85.2** | 85.2 | 84.8 |
| Dropout | **74.9** | **73.3** | **86.0** | **86.3** | 84.0 |
| Logit penalty | **74.6** | **75.9** | 83.5 | 84.8 | 83.3 |
| Logit normalization | **74.2** | **73.6** | **85.6** | 82.2 | 83.8 |
| Cosine softmax | **73.3** | 71.9 | 83.5 | 81.6 | 82.2 |
| Squared error | 71.2 | 67.5 | 76.2 | 73.0 | 75.0 |
| *i.i.d. split from training set (22,431 images):* | | | | | |
| Softmax | **65.8** | **76.2** | **66.8** | **79.9** | **81.4** |
| Label smoothing | 64.9 | 74.9 | 66.3 | 79.2 | 80.2 |
| Sigmoid | 65.4 | 74.9 | **66.5** | 79.3 | 80.5 |
| More final layer $L^2$ | 64.7 | 73.6 | 65.8 | 78.4 | 79.7 |
| Dropout | 65.0 | 74.6 | **66.5** | 79.0 | 80.4 |
| Logit penalty | 64.1 | 72.7 | 65.3 | 78.1 | 78.9 |
| Logit normalization | 63.9 | 72.0 | 65.2 | 77.6 | 78.4 |
| Cosine softmax | 62.6 | 70.0 | 64.2 | 76.4 | 76.7 |
| Squared error | 59.6 | 64.9 | 59.9 | 72.2 | 69.5 |

### D.4 CKA between models before and after transfer

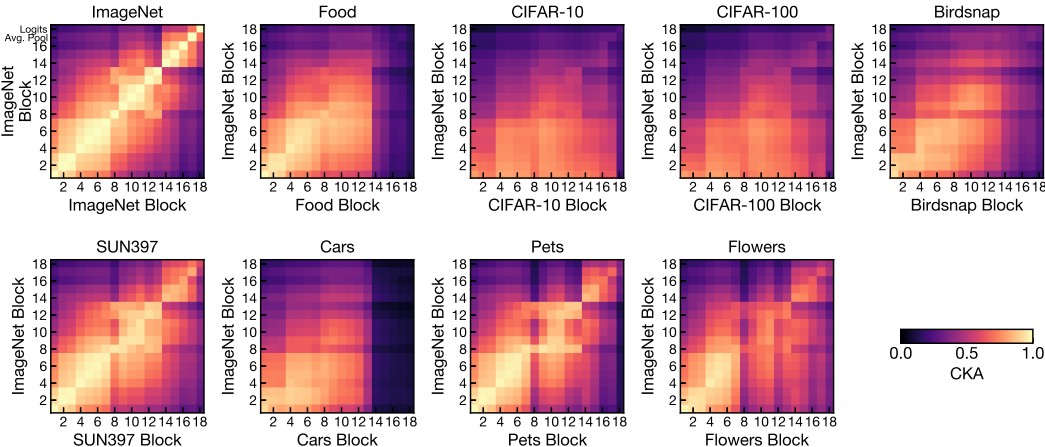

Figure D.3: **Transfer produces large changes in later network layer representations.** We measure CKA between ResNet-50 models trained with softmax cross-entropy before and after transfer, following the same procedure as described in Section 3.2 of the main text. Consistent with previous work, we find that later layers change more than earlier layers, although interestingly, the extent of the changes differs greatly by dataset.

## E  Additional class separation results

### E.1  Relation of class separation index to variance ratios

The class separation index we use is a simple multidimensional generalization of $\eta^2$ in ANOVA or $R^2$ in linear regression with categorical predictors when it is applied to normalized embeddings. Its properties are likely to be familiar to many readers. In this section, for completeness, we derive these properties and provide connections to related work.

The ratio of the average within-class cosine distance to the overall average cosine distance provides a measure of how distributed examples within a class are that is between 0 and 1. We take one minus this quantity to get a closed-form measure of class separation

$$R^2 = 1 - \frac{\sum_{k=1}^{K} \sum_{m=1}^{N_k} \sum_{n=1}^{N_k} \left(1 - \operatorname{sim}(\boldsymbol{X}_{m,:}^{k}, \boldsymbol{X}_{n,:}^{k})\right)/(KN_k^2)}{\sum_{j=1}^{K} \sum_{k=1}^{K} \sum_{m=1}^{N_j} \sum_{n=1}^{N_k} \left(1 - \operatorname{sim}(\boldsymbol{X}_{m,:}^{j}, \boldsymbol{X}_{n,:}^{k})\right)/(K^2 N_j N_k)}, \tag{12}$$

where $N_k$ is the number of examples in class $k$, $\boldsymbol{X}^k \in \mathbb{R}^{N_k \times P}$ is the matrix of $P$-dimensional embeddings of these examples, and $\operatorname{sim}(\boldsymbol{x}, \boldsymbol{y}) = \boldsymbol{x}^{\mathsf{T}} \boldsymbol{y}/(\|\boldsymbol{x}\|\|\boldsymbol{y}\|)$ is cosine similarity.

**Relation of $R^2$ to ratio of within-class vs. total variance:**  $R^2$ is one minus the ratio of the within-class to weighted total variances of $L^2$-normalized embeddings, summed over the feature dimension. To see this, first note that

$$\|\boldsymbol{x}/\|\boldsymbol{x}\| - \boldsymbol{y}/\|\boldsymbol{y}\|\|^2 = (\boldsymbol{x}/\|\boldsymbol{x}\| - \boldsymbol{y}/\|\boldsymbol{y}\|)^{\mathsf{T}} (\boldsymbol{x}/\|\boldsymbol{x}\| - \boldsymbol{y}/\|\boldsymbol{y}\|) \tag{13}$$
$$= 2 - 2\operatorname{sim}(\boldsymbol{x}, \boldsymbol{y}), \tag{14}$$

so, letting $\tilde{\boldsymbol{X}}^k \in \mathbb{R}^{N_k \times P}$ be matrices of $L^2$-normalized embeddings $\tilde{\boldsymbol{X}}_{m,:}^{k} = \boldsymbol{X}_{m,:}^{k}/\|\boldsymbol{X}_{m,:}^{k}\|$

$$R^2 = 1 - \frac{\sum_{k=1}^{K} \sum_{m=1}^{N_k} \sum_{n=1}^{N_k} \|\tilde{\boldsymbol{X}}_{m,:}^{k} - \tilde{\boldsymbol{X}}_{n,:}^{k}\|^2/(KN_K^2)}{\sum_{j=1}^{K} \sum_{k=1}^{K} \sum_{m=1}^{N_j} \sum_{n=1}^{N_k} \|\tilde{\boldsymbol{X}}_{m,:}^{j} - \tilde{\boldsymbol{X}}_{n,:}^{k}\|^2/(K^2 N_j N_k)}. \tag{15}$$

The variance of a vector is a V-statistic with the kernel $h(x, y) = (x - y)^2/2$, i.e.,

$$\operatorname{Var}(\boldsymbol{y}) = \frac{1}{n} \sum_{i=1}^{n} \left(y_i - \sum_{j=1}^{n} y_j/n\right)^2 = \frac{1}{n^2} \sum_{i=1}^{n} \sum_{j=1}^{n} (y_i - y_j)^2/2, \tag{16}$$

and thus the sum of the variances of the columns of a matrix $\boldsymbol{Y} \in \mathbb{R}^{N \times P}$ is

$$\sum_{p=1}^{P} \text{Var}(\boldsymbol{Y}_{:,p}) = \frac{1}{n^2} \sum_{p=1}^{P} \sum_{m=1}^{N} \sum_{n=1}^{N} (Y_{m,p} - Y_{n,p})^2 / 2 = \frac{1}{n^2} \sum_{m=1}^{N} \sum_{n=1}^{N} \|\boldsymbol{Y}_{m,:} - \boldsymbol{Y}_{n,:}\|^2 / 2. \quad (17)$$

If all $N_k$ are equal[5], we can use (17) to write $R^2$ in terms of the ratio of the average within-class variance to the total variance of the normalized embeddings, where each variance is summed over the embedding dimensions:

$$\sigma_{\text{within}}^2 = \sum_{p=1}^{P} \sum_{k=1}^{K} \text{Var}(\tilde{\boldsymbol{X}}_{:,p}^k)/K \qquad \sigma_{\text{total}}^2 = \sum_{p=1}^{P} \text{Var}(\tilde{\boldsymbol{X}}_{:,p}^{\text{all}}) \qquad R^2 = 1 - \frac{\sigma_{\text{within}}^2}{\sigma_{\text{total}}^2}, \quad (18)$$

where $\tilde{\boldsymbol{X}}^{\text{all}} \in \mathbb{R}^{kN \times P}$ is the matrix of all examples.

**Relation of $R^2$ to ratio of between-class vs. total variance:** Letting $\boldsymbol{M} \in K \times P$ be the matrix of mean normalized embeddings of each class $\boldsymbol{M}_{k,:} = \frac{1}{N_k} \sum_{m=1}^{N_k} \tilde{\boldsymbol{X}}_{m,:}^k$, the law of total variance states that the variance of each dimension is the sum of the within-class and between-class variances:

$$\text{Var}(\tilde{\boldsymbol{X}}_{:,p}^{\text{all}}) = \sum_{k=1}^{K} \text{Var}(\tilde{\boldsymbol{X}}_{:,p}^k)/K + \text{Var}(\boldsymbol{M}_{:,p}). \quad (19)$$

Thus, if we let $\sigma_{\text{between}}^2 = \sum_{p=1}^{P} \text{Var}(\boldsymbol{M}_{:,p})$, the variance of the class means summed across dimensions, (19) implies that $\sigma_{\text{total}}^2 = \sigma_{\text{within}}^2 + \sigma_{\text{between}}^2$. Thus, we have

$$R^2 = \sigma_{\text{between}}^2 / \sigma_{\text{total}}^2. \quad (20)$$

**Relation of $R^2$ with other variance ratios:** Other work has used the alternative variance ratios $\sigma_{\text{within}}^2 / \sigma_{\text{between}}^2$ [5], $\sigma_{\text{between}}^2 / \sigma_{\text{within}}^2$ [19], or $(\sigma_{\text{between}}^2 - \sigma_{\text{within}}^2)/(\sigma_{\text{between}}^2 + \sigma_{\text{within}}^2)$ [21] to measure class separation. These ratios are monotonic functions of $R^2$ and can be computed directly from the numbers we provide:

$$\frac{\sigma_{\text{within}}^2}{\sigma_{\text{between}}^2} = \frac{1}{R^2} - 1, \qquad \frac{\sigma_{\text{between}}^2}{\sigma_{\text{within}}^2} = \frac{R^2}{1 - R^2}, \qquad \frac{\sigma_{\text{between}}^2 - \sigma_{\text{within}}^2}{\sigma_{\text{between}}^2 + \sigma_{\text{within}}^2} = 2R^2 - 1. \quad (21)$$

### E.2 Other class separation indexes and measurements

Table E.1: **Comparison of class separation under different distance indexes.** Cosine (mean-subtracted) subtracts the mean of the activations before computing the cosine distance. All results reported for ResNet-50 on the ImageNet training set.

| Loss/regularizer | Cosine | Cosine (mean-subtracted) | Euclidean distance |
|---|---|---|---|
| Softmax | $0.3494 \pm 0.0002$ | $0.3472 \pm 0.0002$ | $0.3366 \pm 0.0002$ |
| Squared error | $0.8452 \pm 0.0002$ | $0.8450 \pm 0.0002$ | $0.8421 \pm 0.0007$ |
| Dropout | $0.4606 \pm 0.0003$ | $0.4559 \pm 0.0002$ | $0.4524 \pm 0.0003$ |
| Label smoothing | $0.4197 \pm 0.0003$ | $0.4124 \pm 0.0004$ | $0.3662 \pm 0.0005$ |
| Extra final layer $L^2$ | $0.5718 \pm 0.0006$ | $0.5629 \pm 0.0005$ | $0.5561 \pm 0.0005$ |
| Logit penalty | $0.6012 \pm 0.0004$ | $0.5950 \pm 0.0004$ | $0.5672 \pm 0.0004$ |
| Logit normalization | $0.5167 \pm 0.0002$ | $0.5157 \pm 0.0002$ | $0.5326 \pm 0.0002$ |
| Cosine softmax | $0.6406 \pm 0.0003$ | $0.6389 \pm 0.0003$ | $0.6406 \pm 0.0003$ |
| Sigmoid | $0.4267 \pm 0.0003$ | $0.4315 \pm 0.0003$ | $0.4272 \pm 0.0003$ |

---

[5]This equivalence also holds for unequal $N_k$ if the variance is replaced by the inverse-frequency-weighted variance.

Table E.2: **Simplex ETF measurements.** Papyan et al. [25] measure various quantities to demonstrate that the representations of neural networks converge to the simplex equiangular tight frame as the training error goes to 0. Collapse to the equiangular tight frame implies both that class separation is maximal (i.e., $R^2 \to 1$) and that class means are maximally distributed. Here we compute the same quantities for networks trained with different losses and optimal early stopping. $\tilde{\mu}_c$ indicates the mean embedding of the $c^{\text{th}}$ of $C = 1000$ total classes after subtracting the global mean across all classes, and $w$ indicates the classifier weights corresponding to the $c^{\text{th}}$ class. $W$ is the matrix of all classifier weights, whereas $M$ is the matrix of all global-mean-subtracted class mean embeddings. $\Sigma_B$ is the covariance matrix of the class means and $\Sigma_W$ is the within-class covariance matrix. $\text{Tr}(\Sigma_W \Sigma_B^\dagger)/C$, where $\Sigma_B^\dagger$ is the Moore-Penrose pseudoinverse of $\Sigma_B$, measures the collapse of within-class variability relative to between-class variability. This quantity reveals the greatest difference between softmax and other losses, and it is also the most related to class separation; for isotropic covariance matrices, $\text{Tr}(\Sigma_W \Sigma_B^\dagger)/C = P/C(1/R^2 - 1)$ where $P$ is the number of penultimate layer features. $\text{Std}_{c,c'}(\text{sim}(\tilde{\mu}_c, \tilde{\mu}_{c'}))$ and $\text{Avg}_{c,c'}|\text{sim}(\tilde{\mu}_c, \tilde{\mu}_{c'}) + \frac{1}{C-1}|$, which measure the discrepancy between the class mean directions and those that would result in the maximal separation between the class means, also differentiate softmax from other losses. Other quantities do not.

| | $\frac{\text{Std}_c(\|\tilde{\mu}_c\|)}{\text{Avg}_c(\|\tilde{\mu}_c\|_2)}$ | $\frac{\text{Std}_c(\|w_c\|)}{\text{Avg}_c(\|w_c\|_2)}$ | $\text{Std}_{c,c'}(\text{sim}(\tilde{\mu}_c, \tilde{\mu}_{c'}))$ | $\text{Std}_{c,c'}(\text{sim}(w_c, w_{c'}))$ |
|---|---|---|---|---|
| Softmax | $0.127 \pm 0.0007$ | $0.100 \pm 0.0005$ | $\mathbf{0.119} \pm 0.0003$ | $0.034 \pm 0.0000$ |
| Label Smoothing | $0.102 \pm 0.0054$ | $0.126 \pm 0.0004$ | $0.096 \pm 0.0010$ | $0.024 \pm 0.0001$ |
| Sigmoid | $0.171 \pm 0.0009$ | $0.102 \pm 0.0003$ | $0.080 \pm 0.0005$ | $0.031 \pm 0.0001$ |
| More $L^2$ | $0.131 \pm 0.0005$ | $0.069 \pm 0.0006$ | $0.086 \pm 0.0003$ | $0.057 \pm 0.0001$ |
| Dropout | $0.134 \pm 0.0012$ | $0.048 \pm 0.0003$ | $0.085 \pm 0.0003$ | $0.061 \pm 0.0001$ |
| Logit Penalty | $0.105 \pm 0.0019$ | $0.092 \pm 0.0002$ | $0.057 \pm 0.0003$ | $0.016 \pm 0.0000$ |
| Logit Norm | $0.300 \pm 0.0014$ | $\mathbf{0.135} \pm 0.0007$ | $0.060 \pm 0.0002$ | $0.020 \pm 0.0000$ |
| Cosine Softmax | $0.100 \pm 0.0004$ | $0.000 \pm 0.0000$ | $0.045 \pm 0.0002$ | $\mathbf{0.063} \pm 0.0001$ |
| Squared Error | $\mathbf{0.448} \pm 0.0050$ | $0.078 \pm 0.0010$ | $0.011 \pm 0.0001$ | $0.005 \pm 0.0001$ |

| | $\text{Avg}_{c,c'}|\text{sim}(\tilde{\mu}_c, \tilde{\mu}_{c'}) + \frac{1}{C-1}|$ | $\text{Avg}_{c,c'}|\text{sim}(w_c, w_{c'}) + \frac{1}{C-1}|$ | $\|W/\|W\|_F - M/\|M\|_F\|_F^2$ | $\text{Tr}(\Sigma_W \Sigma_B^\dagger)/C$ |
|---|---|---|---|---|
| Softmax | $\mathbf{0.084} \pm 0.0003$ | $0.025 \pm 0.0000$ | $0.918 \pm 0.0015$ | $\mathbf{21.158} \pm 0.1283$ |
| Label Smoothing | $0.068 \pm 0.0008$ | $0.018 \pm 0.0001$ | $0.954 \pm 0.0052$ | $9.001 \pm 0.0859$ |
| Sigmoid | $0.057 \pm 0.0005$ | $\mathbf{0.056} \pm 0.0004$ | $\mathbf{0.960} \pm 0.0013$ | $6.957 \pm 0.0371$ |
| More $L^2$ | $0.059 \pm 0.0003$ | $0.042 \pm 0.0001$ | $0.328 \pm 0.0009$ | $6.104 \pm 0.1007$ |
| Dropout | $0.057 \pm 0.0003$ | $0.045 \pm 0.0001$ | $0.521 \pm 0.0010$ | $9.701 \pm 0.0494$ |
| Logit Penalty | $0.034 \pm 0.0003$ | $0.010 \pm 0.0000$ | $0.481 \pm 0.0020$ | $2.722 \pm 0.2005$ |
| Logit Norm | $0.034 \pm 0.0002$ | $0.013 \pm 0.0000$ | $0.759 \pm 0.0016$ | $2.943 \pm 0.1150$ |
| Cosine Softmax | $0.023 \pm 0.0002$ | $0.045 \pm 0.0001$ | $0.302 \pm 0.0004$ | $1.305 \pm 0.3153$ |
| Squared Error | $0.002 \pm 0.0000$ | $0.002 \pm 0.0000$ | $0.647 \pm 0.0043$ | $0.334 \pm 0.0260$ |

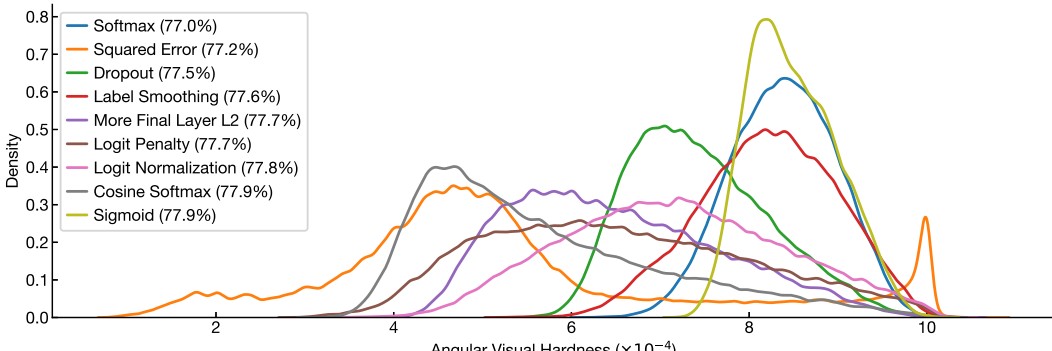

Figure E.1: **Angular visual hardness of different loss functions.** Kernel density estimate of the angular visual hardness [3] scores of the 50,000 examples in the ImageNet validation set, computed with a Gaussian kernel of bandwidth $5 \times 10^{-6}$, for ResNet-50 networks trained with different losses. Legend shows ImageNet top-1 accuracy for each loss function in parentheses. Although alternative loss functions generally reduce angular visual hardness vs. softmax loss, sigmoid loss does not, yet it is tied for the highest accuracy of any loss function.

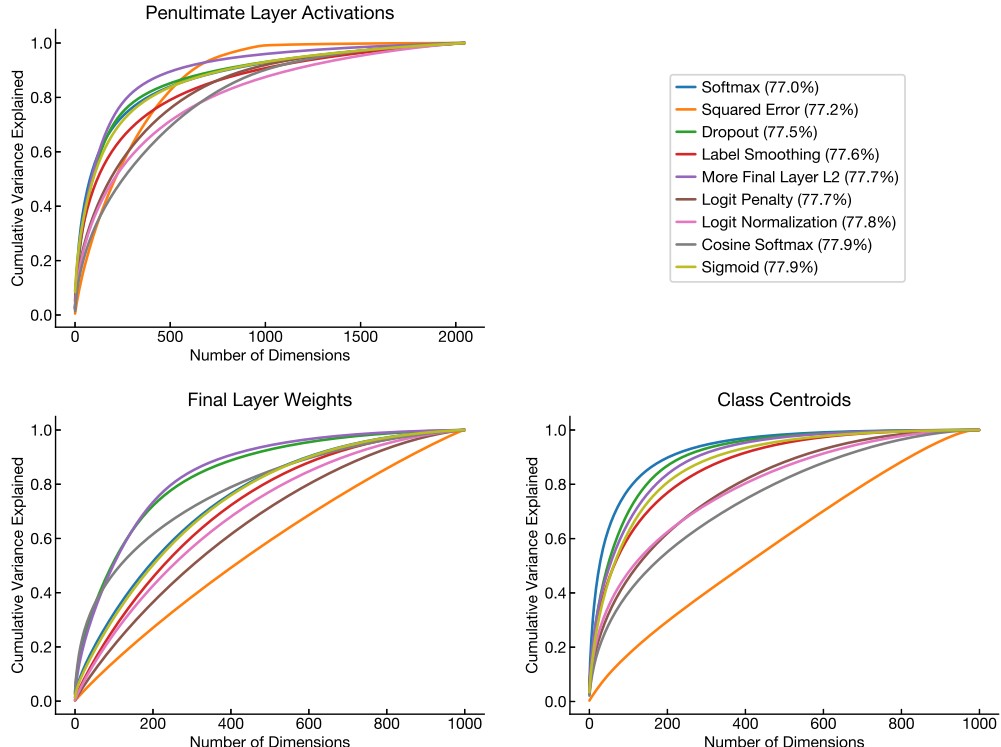

Figure E.2: **Singular value spectra of activations and weights learned by different losses.** Singular value spectra computed for penultimate layer activations, final layer weights, and class centroids of ResNet-50 models on the ImageNet training set. Penultimate layer activations and final layer weights fail to differentiate sigmoid cross-entropy from softmax cross-entropy. By contrast, the singular value spectrum of the class centroids clearly distinguishes these losses.

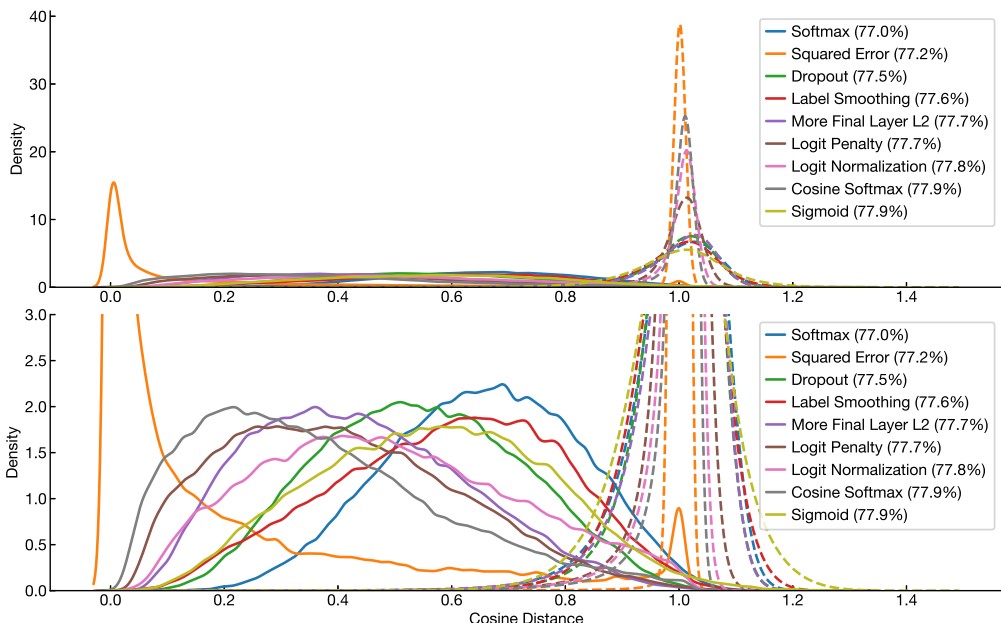

Figure E.3: **The distribution of cosine distance between examples.** Kernel density estimate of the cosine distance between examples of the same class (solid lines) and of different classes (dashed lines), for penultimate layer embeddings of 10,000 training set examples from ResNet-50 on ImageNet. Top and bottom plots show the same data with different y scales.

### E.3   Augmentation can improve accuracy without increasing class separation

In Section C.4, we show that improved loss functions and AutoAugment are additive, whereas combinations of improved loss functions or regularizers lead to no significant accuracy improvements. In Table E.3 below, we show that AutoAugment also does not increase class separation. These results confirm that data augmentation and modifications to networks' final layers exert their effects via different (and complementary) mechanisms.

Table E.3: **AutoAugment increases ImageNet top-1 accuracy without increasing class separation.** Top-1 accuracy is computed on the ImageNet validation set; class separation is computed on the ImageNet training set. Results are averaged over 3 (with AutoAugment) or 8 (standard augmentation) models.

| | Standard augmentation | | AutoAugment | |
|---|---|---|---|---|
| Loss | ImageNet top-1 | Class sep. ($R^2$) | ImageNet top-1 | Class sep. ($R^2$) |
| Softmax | $77.0 \pm 0.06$ | $0.349 \pm 0.0002$ | $77.7 \pm 0.05$ | $0.353 \pm 0.0002$ |
| Sigmoid | $77.9 \pm 0.05$ | $0.427 \pm 0.0003$ | $78.5 \pm 0.04$ | $0.432 \pm 0.0001$ |
| Logit penalty | $77.7 \pm 0.04$ | $0.601 \pm 0.0004$ | $78.3 \pm 0.05$ | $0.595 \pm 0.0003$ |
| Cosine softmax | $77.9 \pm 0.02$ | $0.641 \pm 0.0003$ | $78.3 \pm 0.05$ | $0.632 \pm 0.0001$ |

### E.4   Training dynamics of class separation

As we discuss in detail in Section 3.3 of the main text, different loss functions lead to different values of class separation. However, we train models with different loss functions for different numbers of epochs, reported in Table A.1. In this section, we confirm that differences in the number of training epochs alone do not explain differences in observed class separation among losses. Instead, as shown in Figure E.4, differences among losses are established early in training and the relative ordering changes little. We observe that, for the softmax cross-entropy model, class separation peaks at epoch 32 and then falls, whereas all models trained with different objectives achieve maximum class separation on the training set at the last checkpoint. On the validation set, for most losses, class separation peaks before optimal accuracy is reached.

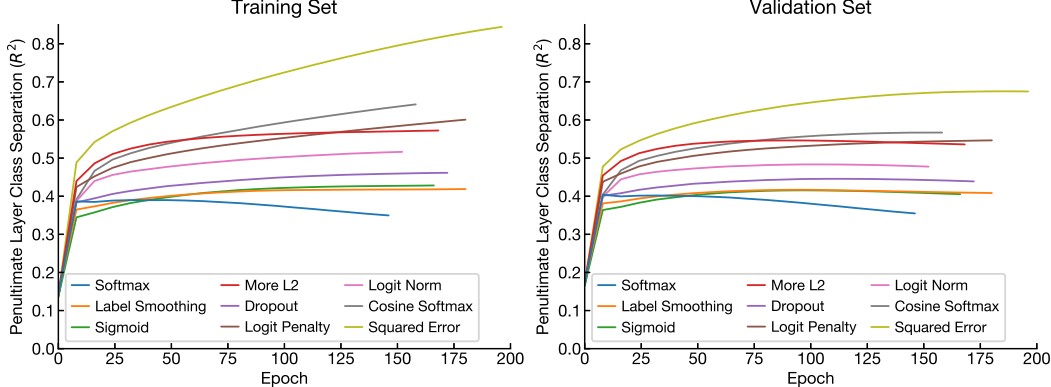

Figure E.4: **Differences in training dynamics of class separation across loss functions.** Plots show evolution of class separation over training on the ImageNet training set (left) and validation set (right), for a single ResNet-50 model of each loss function type evaluated every 8 epochs. Curves terminate at the epoch that provided the highest holdout set accuracy.

## F   Class overlap between ImageNet and transfer datasets

In this section, we investigate overlap in the classes contained in the ImageNet ILSVRC 2012 dataset and those contained in the downstream datasets investigated in this work. We previously reported the overlap in the *images* contained in these datasets and those contained in ImageNet in Appendix H of Kornblith et al. [16].

We first measure the number of classes in each dataset where the class names correspond semantically to classes in ImageNet. Due to differences in the granularity of different datasets, semantic class overlap is somewhat ambiguous. For example, CIFAR-10 contains a single "dog" class that corresponds to 90 dog breeds contained in ImageNet, but ImageNet contains only a single "hummingbird" class whereas Birdsnap contains 9 different species. We consider classes as "overlapping" when the name of the downstream either directly or nearly corresponds to an ImageNet class, or is a superclass of ImageNet classes.

In addition to being ambiguous, semantic class overlap does not consider shift in class-conditional distributions. Simply because classes in two datasets refer to the same kinds of real-world objects does not mean that the images those classes contain are similar. For example, 61 of the 100 classes in CIFAR-100 are superclasses of ImageNet classes, but because CIFAR images are much lower resolution, a classifier trained on ImageNet does not perform well at classifying them.

To develop a measure of class overlap that takes distribution shift into consideration, we map each ImageNet class to a class in the downstream dataset and use this mapping in combination with the original 1000-way ImageNet-trained vanilla softmax cross-entropy network to measure classification accuracy. Finding the optimal class mapping is an instance of the minimum-cost flow problem, but can also be solved somewhat less efficiently as a variant of the assignment problem. For each downstream task, we apply an ImageNet classifier to the task's training set and compute the matching matrix. The cost matrix for the assignment problem is the negative matching matrix. To allow multiple ImageNet classes to be assigned to the same downstream task class, we replicate all classes in the downstream task $k$ times, where we select $k$ so that $k+1$ replications provides no improvement in accuracy, then use `scipy.optimize.linear_sum_assignment` to find the mapping. We call the accuracy of the resulting mapping the "assignment accuracy."

Results are shown in Table F.1. Semantic class overlap is generally low, but CIFAR-10, CIFAR-100, and Pets all have non-trivial semantic class overlap. For most datasets, the assignment accuracy is less than half the linear transfer accuracy; the only exceptions are CIFAR-10 and Pets, where the drop is smaller. Pets has comparable linear transfer accuracy to CIFAR-10 but higher assignment accuracy, and thus arguably has the greatest class overlap with ImageNet.

Table F.1: **Class overlap between ImageNet and transfer datasets.**

| Dataset | Number of classes | Semantic class overlap | Assignment accuracy | Linear transfer accuracy |
|---|---|---|---|---|
| Food | 101 | 7 | 15.4 | 74.6 |
| CIFAR-10 | 10 | 9 | 65.1 | 92.4 |
| CIFAR-100 | 100 | 61 | 34.6 | 76.9 |
| Birdsnap | 500 | 11 | 7.0 | 55.4 |
| SUN397 | 397 | 39 | 20.2 | 62.0 |
| Cars | 196 | 0 | 4.0 | 60.3 |
| Pets | 37 | 25 | 71.2 | 92.0 |
| Flowers | 102 | 1 | 7.6 | 94.0 |