# OpenReview forum: "Why Do Better Loss Functions Lead to Less Transferable Features?"
_NeurIPS.cc/2021/Conference — NeurIPS 2021 Poster_

### Official Review · Reviewer_xVq5 · 2021-07-11

**Rating:** 7
**Confidence:** 4

**Summary:**

The paper studies the role of downstream performance on the given task and its role on transferring to a novel data distribution. The paper rigorously studies the role of different loss functions and its effect on transfer learning (with or without finetuning) while also investigating other learned features of the NN. The paper concludes that learning better downstream discriminative features may not lead to better transfer learning models.

**Limitations And Societal Impact:**

The paper is an experimental study in transfer learning and therefore may have societal impact in the form of harmful features being learned during training. The paper does not discuss this directly, however the experimental study is somewhat orthogonal to the societal impact.

**Main Review:**

The paper is well studied and a clear investigation into learning transferrable features and how learning better downstream classifiers might lead to slightly worsening performance of the models. The study investigates various properties of the learned network such as sparsity, early layer representations, class separation over 7 different chosen loss function and variants. The paper delivers a very clear message despite being limited in some ways such as:

- experiments with non-resnet CNNs
- experiments with non-CNN models, such as VIT
- analysis of similar papers (for example big transfer)
- the role of self-supervised pretraining and if better self-supervised models lead to better transfer over all domains (but that might be a whole research project in itself)

The main role of Figure 2 is also a bit confusing to me, as in what is it trying to show. If I understand the plot correctly, is it suggesting that similar features are learned despite the choice of the loss function? Please correct me if I am mistaken. In lines 167-170 the paper suggests that there are a lot of similarities in the early layers but not as many in the later layers, but I am unsure if that is the case since the diagonal appears to be somewhat similar except in certain cases (softmax vs squared error for example), and the differences in the later layers are somewhat expected.

The paper is missing a couple citations in the form of [1] which reaches a somewhat different conclusion to the paper and [2] which also talks about how tightly the clustering is in the representation space and how tighter clusters hurt the downstream transfer properties (very similar to the class separability conclusions). It might be also useful for the paper to separate itself more from Kornblith et al. 2018.

[1] http://proceedings.mlr.press/v97/recht19a.html
[2] https://arxiv.org/abs/2002.08473

**Time Spent Reviewing:**

1.5 hours

---

> ### Author Response · Authors · 2021-08-10
> **Response to Reviewer Comments**
>
> We thank the reviewer for their helpful comments. We address each below.
>
> **Experiments with non-ResNet CNNs**
>
> Since several reviewers have asked for similar experiments, we have replicated our findings with Inception v3, which has no residual connections. For each loss, we trained 3 models, using the same loss function hyperparameters as for ResNet-50. We cannot show the representational similarity plots here, but they are qualitatively similar to those shown in the paper. The patterns we observe in the transfer accuracies as well as the penultimate layer class separation are consistent with those for ResNet-50. We present the results in a table below. As in the paper, results not statistically significantly different from the best are bolded.
>
> |                     | ImageNet Top-1 | Top-5      | Class Sep. ($R^2$) | Food   | CIFAR10   | CIFAR100   | Birdsnap   | SUN397   | Cars   | Pets   | Flowers   |
> |:--------------------|:---------------|:-----------|:---------------------|:-------|:----------|:-----------|:-----------|:---------|:-------|:-------|:----------|
> | Softmax             | 78.6           | 94.24      | 0.356                    | **74.5** | **92.4**    | **76.2**     | **59.3**     | **63.1**   | **64.4** | 92.2   | **94.0**    |
> | Label Smoothing     | 78.8           | **94.60**  | 0.441                    | 73.3   | 91.6      | **75.0**     | 56.1       | 62.4     | 60.3   | **93.0** | 92.4      |
> | Sigmoid             | **79.1**       | 94.17      | 0.444                    | 73.7   | 91.3      | 74.7       | 55.0       | 62.0     | 60.7   | **92.8** | *93.0*    |
> | More Final Layer L2 | **79.0**       | 94.52      | 0.586                    | 70.1   | 91.0      | 73.3       | 52.4       | 61.0     | 51.1   | **92.5** | 89.6      |
> | Dropout             | **79.0**       | **94.50**  | 0.454                    | 72.6   | 91.5      | 74.7       | 56.3       | 62.1     | 59.2   | **92.7** | 92.2      |
> | Logit Penalty       | **78.9**       | **94.63**  | 0.638                    | 69.1   | 90.6      | 72.1       | 49.3       | 59.2     | 52.3   | 92.3   | 87.9      |
> | Logit Normalization | **78.8**       | 94.34      | 0.559                    | 67.4   | 90.6      | 72.2       | 50.9       | 58.5     | 45.6   | 92.1   | 84.2      |
> | Cosine Softmax      | **78.9**       | 94.38      | 0.666                    | 63.1   | 90.3      | 71.5       | 45.8       | 55.6     | 38.0   | 90.6   | 75.2      |
> | Squared Error       | 77.7           | 93.28      | 0.838                    | 45.3   | 84.1      | 57.6       | 25.0       | 41.1     | 18.8   | 85.7   | 54.8      |
>
> **Experiments with non-CNN models**
>
> Based on the ResNet-50 and Inception v3 results, we suspect that our results hold independent of the network backbone. We acknowledge that additional confirmation could be of value, but these experiments are expensive to perform, and we were unable to do so during the rebuttal period.
>
> **Analysis of similar papers (for example big transfer)**
>
> To our knowledge, Big Transfer does not manipulate the loss function, but instead manipulates training data and model capacity. Further investigation of these properties is orthogonal to the current work’s focus on the role of loss functions, and would demand at least slightly different analysis techniques. For example, we would not expect the exact values of class separation to be comparable across datasets.
>
> **Analysis of self-supervised models**
>
> >the role of self-supervised pretraining and if better self-supervised models lead to better transfer over all domains (but that might be a whole research project in itself)
>
> We believe this is a topic worthy of investigation, and we are aware of some previous work in this area (e.g. [1,2,3]). However, as the reviewer acknowledges, investigation of self-supervised learning would involve substantially expanding the scope of the current paper.
>
> **Importance of Figure 2**
>
> >The main role of Figure 2 is also a bit confusing to me, as in what is it trying to show.
>
> The focus of this section is investigating which layers of neural networks change when one swaps out vanilla softmax cross-entropy for a different loss function. The panels of Figure 2(a) measure similarity between all pairs of blocks from two networks trained with the same or different loss functions. It's true that the diagonals in Figure 2(a) are typically substantially stronger than the off-diagonals, indicating that a layer in one net is more similar to the architecturally corresponding layer in the other net than to other layers regardless of which loss functions they were trained with. However, in the last few layers of the network, the diagonals are substantially brighter for the softmax vs. softmax comparison than for comparisons between softmax-trained networks and other networks. This is perhaps easier to see in the panels of Figure 2(b), which compare representations learned by all loss functions at a single block. In these heatmaps, it’s clear that differences among networks trained with different loss functions first become apparent around block 13 (out of 16 total ResNet blocks).
>
> **Additional related work**
>
> We agree that the work the reviewer points to is relevant, and we will discuss it in the paper.
>
> **References**
>
> [1] Ericsson, L., Gouk, H., & Hospedales, T. M. (2021). How Well Do Self-Supervised Models Transfer?. In Proceedings of the IEEE/CVF Conference on Computer Vision and Pattern Recognition (pp. 5414-5423).
>
> [2] Islam, A., Chen, C. F., Panda, R., Karlinsky, L., Radke, R., & Feris, R. (2021). A Broad Study on the Transferability of Visual Representations with Contrastive Learning. arXiv preprint arXiv:2103.13517.
>
> [3] Kotar, K., Ilharco, G., Schmidt, L., Ehsani, K., & Mottaghi, R. (2021). Contrasting Contrastive Self-Supervised Representation Learning Models. arXiv preprint arXiv:2103.14005.

---

> > ### Comment · Reviewer_xVq5 · 2021-08-11
> > **Happy with the rebuttal**
> >
> > Thanks for the comments. I have updated the score to a 7. I do recommend the authors to consider adding some (even one) ViT / MLP Mixer type architecture into their suite of experiments, as they did with the Inception networks.

---

### Official Review · Reviewer_MRUN · 2021-07-12

**Rating:** 7
**Confidence:** 4

**Summary:**

The authors study how different loss functions and regularizers affect the learned representation of a ResNet-50. They find that although sigmoid, label smoothing, etc. can improve ImageNet performance, the learned representations do not transfer as well as when training with softmax cross-entropy. The main hypothesis is that softmax attains embeddings with lower inter-class separation and higher intra-class variance, which is verified empirically by computing the $\frac{d_{intraclass}}{d_{total}}$. In addition, they also use centered kernel alignment (CKA) to compare representations learned with different loss functions at different depths of the neural networks. They observe that most of the representational differences between models trained with different loss functions happen closer to the output. As a result, fine-tuning the models on new data is enough to eliminate the performance discrepancies observed when linear probing.

**Limitations And Societal Impact:**

The authors have addressed the limitations but not the negative societal impact.

**Main Review:**

# Overall Review
I found this study interesting and necessary in order to better understand how different design choices affect the performance of deep learning models. The text is well-written, the experiments are sound, and the conclusions are instructive. I have some questions regarding the experimental setup (see below), particularly on the choice of loss functions and optimizer. For the moment I think that the strengths outweigh the weaknesses of this work and thus I recommend its acceptance.

## Strengths
* Understanding how the choice of loss functions affect the learning of deep representations is an important topic.
* The authors explore the problem from different perspectives like CKA, class separation, simplex ETF measurements, angular visual hardness, and representation sparsity.
* The text is clear and well-written.

## Weaknesses
* The choice of optimizer and loss function is not clear (see detailed comments).
* Transfer results are only shown from ImageNet to other datasets. It would have been interesting to see what happens when trying to transfer from another dataset (see detailed comments).


# Detailed comments
## Originality
* To the best of my knowledge this is the first work exploring transferability of different pre-existing loss functions. There are previous works on meta-larning more transferable functions [B, C], previous studies on different loss functions [D], and specific loss functions that address the intraclass compactness problem [E] but they are orthogonal to the work presented in this paper.

[B] Li, Chuming, et al. "Am-lfs: Automl for loss function search." Proceedings of the IEEE/CVF International Conference on Computer Vision. 2019.

[C] Huang, Chen, et al. "Addressing the loss-metric mismatch with adaptive loss alignment." International Conference on Machine Learning. PMLR, 2019.

[D] Janocha, Katarzyna, and Wojciech Marian Czarnecki. "On Loss Functions for Deep Neural Networks in Classification." Schedae Informaticae 25 (2016).

[E] Luo, Yan, et al. "$\mathcal {G} $-softmax: improving intraclass compactness and interclass separability of features." IEEE transactions on neural networks and learning systems 31.2 (2019): 685-699.

## Quality
* The overall quality is good, in particular on the experimental part with with experiments on 9 different loss functions over 8 different datasets (+ImageNet versions), and the evaluation over multiple metrics. For instance, given the current interest in self-supervised learning, it would have been interesting to try a contrastive loss such as the one described in [F]
* However it is not clear why you chose those particular loss functions. For instance, benchmarking a meta-learned loss function could have provided more insight on whether meta-learning-based loss functions overfit the meta-training set.  Could you comment on this point?
* SGD is sensitive to the magnitude of the loss, thus, different loss functions should have a different learning rate. Do you account for this fact? One way to alleviate this problem is using Adam.

[F] Khosla, Prannay, et al. "Supervised Contrastive Learning." Advances in Neural Information Processing Systems 33 (2020).

## Clarity
* In general, the text is well-written and easy to follow.
* There is a minor typo in the supplementary material in the caption of Table B3 ("see Table ??").

## Significance
* The proposed study is sound and interesting for the research community. The experiments are extensive and insightful.


**Time Spent Reviewing:**

4

---

> ### Author Response · Authors · 2021-08-10
> **Response to Reviewer Comments**
>
> We thank the reviewer for their very useful comments. We address their points below.
>
> **Importance of optimizer and loss scale**
>
> >The choice of optimizer and loss function is not clear...SGD is sensitive to the magnitude of the loss, thus, different loss functions should have a different learning rate.
>
> We use SGD + momentum because it is the optimizer that is most commonly used to train ResNets on ImageNet. It is true that SGD is sensitive to the magnitude of the gradient. Where possible, we attempt to match the scale of the gradients with respect to the correct class logits for all loss functions, which should lead to similar overall gradient scales. We conducted some pilot experiments where we varied the loss scale, and found that manipulating it made little difference to overall accuracy. Despite using optimization hyperparameters that were originally tuned for vanilla softmax cross-entropy, we were still able to achieve better accuracy using each of the investigated alternative loss functions.
>
> There are reasons to expect that the effect of the loss scale should be minor. SGD training quickly reaches an equilibrium where the magnitude of the update related to the loss gradient approximately equals the amount by which the weights are decayed [1,2,3]. The “effective learning rate” (the ratio between the norms of the gradients and weights) at equilibrium is determined by the _product_ of the learning rate and weight decay. Because networks that incorporate normalization layers are invariant to the scale of the activations preceding those layers and thus to the scale of the weights, the effective learning rate summarizes the magnitude of the functional change. Rescaling the loss is equivalent to increasing the learning rate while decreasing the weight decay, and does not change the effective learning rate. We thus expect that the scale of the loss should have little effect as long as it does not disrupt the early phase of training or the training of non-normalized parameters.
>
> We acknowledge that further hyperparameter tuning could lead to small improvements in pretraining accuracy, but accuracy on the pretraining task is not the primary focus of our investigation. Moreover, given that we have to perform 3 ImageNet training runs for each hyperparameter configuration in order to reliably identify optimal hyperparameters, adding additional dimensions to our hyperparameter grid is computationally infeasible.
>
> **Transfer from other datasets**
>
> >Transfer results are only shown from ImageNet to other datasets. It would have been interesting to see what happens when trying to transfer from another dataset (see detailed comments).
>
> We agree that it would be interesting to conduct a similar exploration of the effect of the dataset upon representations and transfer. However, further investigation of the role of the dataset is orthogonal to our focus on the role of loss functions, and would demand different analysis techniques. For example, we would not expect the exact values of class separation to be comparable across datasets.
>
> **Contrastive and meta-learned losses**
>
> >it would have been interesting to try a contrastive loss...benchmarking a meta-learned loss function could have provided more insight on whether meta-learning-based loss functions overfit the meta-training set
>
> We acknowledge that an enormous amount of effort has been put into designing loss functions for deep learning, and our experiments only cover a small number of the methods that have been proposed. In particular, we excluded loss functions that require specially constructed batches, including supervised contrastive learning. It could be interesting to compare the accuracy of meta-learned loss functions with optimally tuned single-parameter loss functions. However, we see no reason to believe that our general findings would change. Since meta-learned loss functions are generally optimized for accuracy on the pretraining task, and not for transfer accuracy, they are likely to encounter the same tradeoff that we observe here.
>
> **References**
>
> [1] Page, D. (2018). How to train your ResNet. https://myrtle.ai/learn/how-to-train-your-resnet-6-weight-decay/
>
> [2] Chiley, V., Sharapov, I., Kosson, A., Koster, U., Reece, R., Samaniego de la Fuente, S., ... & James, M. (2019). Online normalization for training neural networks. Advances in Neural Information Processing Systems, 32, 8433-8443.
>
> [3] Li, Z., & Arora, S. (2019). An Exponential Learning Rate Schedule for Deep Learning. In International Conference on Learning Representations.

---

> > ### Comment · Reviewer_MRUN · 2021-08-25
> > **Post-rebuttal response**
> >
> > Thanks for your responses! Although they are conservative, with no experiments on Adam nor contrastive losses, I understand it is not easy to provide extra experiments during the time constraints of a rebuttal.
> >
> > I still think your work is interesting and necessary for the research community, so I am inclined to keep the original score of 7 subject to the discussion with the other reviewers.

---

### Official Review · Reviewer_qwG6 · 2021-07-15

**Rating:** 9
**Confidence:** 4

**Summary:**

The development of new loss functions and regularizers is a major domain of ML research, but researchers often prioritize accuracy on the original dataset over transfer learning. The authors of the present work wanted to understand the effects of different training objectives on transferability in image classification tasks. They find that many loss functions outperform the traditional softmax cross-entropy but transfer worse, and provide insight into this phenomenon by examining hidden layer representations: higher original-dataset accuracy collapses within-class variability in deeper layer representations, leading to less generalizable features.

**Limitations And Societal Impact:**

The authors adequately address the limitations of their work in the space provided.

**Main Review:**

The authors comprehensively characterize the effects of different loss functions empirically, not just on transfer learning, but also many performance metrics of interest including accuracy, out-of-distribution/naturalistic robustness, and calibration. I am also convinced by their analysis-based approach to understanding the mechanisms underlying the observed effects of different loss functions on different performance metrics. The work is broadly relevant beyond simply understanding and improving transfer learning. And while I would like to see this work expanded beyond the ResNet50 architecture, I understand that replicating the entirety of this work on larger models would be very resource-intensive.

This is a very thorough paper.

I have a few minor comments:

Optimal validation accuracy is typically achieved after 140-200 epochs of training, which leads to the question of whether the results are consistent across the duration of training. The answer to this is interesting from a scientific perspective, but there is also a practical consideration: it would be very helpful for other ML practitioners to know whether they can expect to see benefits from alternative loss functions at shorter training durations. At the very least, I think the authors should include a plot of validation accuracy as a function of epoch for the different loss functions.

I think it would be helpful to have a version of Figure 1 with the full transfer learning results and/or some measure of variability across different transfer datasets.

The authors state “The relationship between class separation and pretraining accuracy is non-monotonic; squared error achieves the highest class separation but does not significantly outperform vanilla softmax cross-entropy.” I would appreciate seeing some discussion (and ideally analysis) of the mechanisms by which accuracy can be improved other than by increasing class separation in the penultimate layer. Also, perhaps it was cut for space, but there is no motivation provided for why the authors chose to examine class separation in the penultimate layer. It seems like a perfectly sensible property to examine, but the transition to this analysis feels a bit abrupt.

The manuscript could be enriched with some discussion about the very interesting result that none of the loss functions have worse original-dataset accuracy or better transfer accuracy than softmax-cross-entropy, though I understand space is at a premium.


**Time Spent Reviewing:**

4 hours

---

> ### Author Response · Authors · 2021-08-10
> **Response to Reviewer Comments**
>
> We thank the reviewer for their very useful feedback. We address each comment below.
>
> **Other architectures**
>
> >And while I would like to see this work expanded beyond the ResNet50 architecture, I understand that replicating the entirety of this work on larger models would be very resource-intensive.
>
> Since multiple reviewers requested that we investigate the generality of our results across architectures, we have performed similar experiments with Inception v3, which is not exactly a larger model but helps to demonstrate the generality of our results. For each loss, we trained 3 models, using the same loss function hyperparameters as for ResNet-50. We cannot show the representational similarity plots here, but they are qualitatively similar to those shown in the paper. The patterns we observe in the transfer accuracies as well as the penultimate layer class separation are consistent with those for ResNet-50. We present the results in a table below. As in the paper, results not statistically significantly different from the best are bolded.
>
> |                     | ImageNet Top-1 | Top-5      | Class Sep. ($R^2$) | Food   | CIFAR10   | CIFAR100   | Birdsnap   | SUN397   | Cars   | Pets   | Flowers   |
> |:--------------------|:---------------|:-----------|:---------------------|:-------|:----------|:-----------|:-----------|:---------|:-------|:-------|:----------|
> | Softmax             | 78.6           | 94.24      | 0.356                    | **74.5** | **92.4**    | **76.2**     | **59.3**     | **63.1**   | **64.4** | 92.2   | **94.0**    |
> | Label Smoothing     | 78.8           | **94.60**  | 0.441                    | 73.3   | 91.6      | **75.0**     | 56.1       | 62.4     | 60.3   | **93.0** | 92.4      |
> | Sigmoid             | **79.1**       | 94.17      | 0.444                    | 73.7   | 91.3      | 74.7       | 55.0       | 62.0     | 60.7   | **92.8** | *93.0*    |
> | More Final Layer L2 | **79.0**       | 94.52      | 0.586                    | 70.1   | 91.0      | 73.3       | 52.4       | 61.0     | 51.1   | **92.5** | 89.6      |
> | Dropout             | **79.0**       | **94.50**  | 0.454                    | 72.6   | 91.5      | 74.7       | 56.3       | 62.1     | 59.2   | **92.7** | 92.2      |
> | Logit Penalty       | **78.9**       | **94.63**  | 0.638                    | 69.1   | 90.6      | 72.1       | 49.3       | 59.2     | 52.3   | 92.3   | 87.9      |
> | Logit Normalization | **78.8**       | 94.34      | 0.559                    | 67.4   | 90.6      | 72.2       | 50.9       | 58.5     | 45.6   | 92.1   | 84.2      |
> | Cosine Softmax      | **78.9**       | 94.38      | 0.666                    | 63.1   | 90.3      | 71.5       | 45.8       | 55.6     | 38.0   | 90.6   | 75.2      |
> | Squared Error       | 77.7           | 93.28      | 0.838                    | 45.3   | 84.1      | 57.6       | 25.0       | 41.1     | 18.8   | 85.7   | 54.8      |
>
> **Effect of training duration**
>
> >it would be very helpful for other ML practitioners to know whether they can expect to see benefits from alternative loss functions at shorter training durations
>
> Although we have not focused on this topic in the text, there are indeed differences between the training curves of different loss functions. We cannot include images in our response here, but we will add this figure to the Supplementary Material.
>
> **Expanded version of Figure 1**
>
> >I think it would be helpful to have a version of Figure 1 with the full transfer learning results and/or some measure of variability across different transfer datasets.
>
> We will create a version of Figure 1 with separate subpanels for each dataset and add it to the Supplementary Material. We’ll also see whether we can add error bars around the points in the version of Figure 1 that is currently in the main text.
>
> **Improving accuracy without increasing class separation**
>
> >I would appreciate seeing some discussion (and ideally analysis) of the mechanisms by which accuracy can be improved other than by increasing class separation in the penultimate layer.
>
> One such mechanism is data augmentation. In Section D of the supplementary material, we show that combining loss functions with each other generally doesn’t improve accuracy but combining improved loss functions with improved data augmentation does. Below, we provide ImageNet top-1 accuracy and class separation numbers for models trained with a subset of loss functions with standard augmentation and with AutoAugment. AutoAugment improves accuracy but has little effect upon class separation.
>
> | Loss           | Standard Acc.       | Standard Class Sep. | AutoAugment Acc. | AutoAugment Class Sep. |
> |:---------------|:--------------------|:--------------------|:-----------------|:-----------------------|
> | Softmax        | 77.0                | 0.345               | 77.7             | 0.353                  |
> | Sigmoid        | 77.9                | 0.427               | 78.5             | 0.432                  |
> | Logit penalty  | 77.7                | 0.601               | 78.3             | 0.595                  |
> | Cosine softmax | 77.9                | 0.641               | 78.3             | 0.632                  |
>
> **Motivation for investigating class separation**
>
> >Also, perhaps it was cut for space, but there is no motivation provided for why the authors chose to examine class separation in the penultimate layer. It seems like a perfectly sensible property to examine, but the transition to this analysis feels a bit abrupt.
>
> We agree that the transition here could be improved. We proceed from investigating CKA between networks to examining the penultimate layer because we see large differences among loss functions there in the CKA plots toward the end of the networks and the penultimate layer, unlike the logits layer, it is not intrinsically specialized for the specific loss.

---

> > ### Comment · Reviewer_qwG6 · 2021-08-20
> > **Satisfying rebuttal**
> >
> > My only concern regarding the reliability of the findings was that the authors only examined one model architecture. I am happy to see their results extended to Inception v3. My two issues regarding clarity are satisfied. My remaining comments were mostly motivated by curiosity, which the authors graciously indulged. I originally rated this paper a 9 because I thought it was excellent, and my rating will remain a 9 because I still think it's excellent.

---

### Official Review · Reviewer_a9W9 · 2021-07-15

**Rating:** 5
**Confidence:** 4

**Summary:**

This paper studies the impact of common supervised objectives and regularization on transfer learning. With extensive experiments, the authors find out that although these objectives (or regularizers) lead to better performance on the pre-training dataset, their performance on the downstream tasks are always worse. This complements the results in Kornblith et al.. Furthermore, the authors try to probe into the reason of this phenomenon, and find out that these objectives induce better class separation, which is associated with less transferable features.

**Limitations And Societal Impact:**

The authors adequately addressed the limitations and potential negative societal impact

**Main Review:**

This paper studies a very important problem in transfer learning. The manuscript is well written. The claims are stated clearly in each section and supported with thorough experiments. However, I believe several aspects of the manuscript can be improved in its current form.

1)	Although the conclusion of the paper is interesting and solid, I can be limited to the current setup. When the pretraining and target tasks are very similar (in extreme terms, the same), better performance on the pretraining task can lead to better transfer learning. As the similarity decreases, the performance of the pretraining and target tasks are less correlated, and we may enter the regime of the current paper.
2)	The findings of section 3.2 is not surprising as it is common sense that neural networks tend to learn similar “low-level” features in lower layers, and lower layers are more important in fine-tuning. In addition, the accuracies of different objectives vary only by ~1%, which can be too small for an informative CKA visualization.
3)	As to the class separation, many pre-training tasks are not supervised classification, which do not involve it. Even if greater class separation is associated with less transferable features, it is hard to say class separation determines transferability, which cast doubt on the value of this claim.

Post-rebuttal:
After reading the rebuttal and other reviews carefully, I found out some of my concerns addressed. On the other hand, I still believe some part of the experiments and the conclusion of the paper are limited:
1. The downstream tasks are limited to classification. It is fine to me to study only the computer vision setup, but it is necessary to include other type of downstream tasks such as object detection. Even for classification tasks, there exists datasets which are very different from those chosen in this paper. I don't think the 8 datasets are representative of downstream tasks in practical use.
2. As to the concrete value of the paper, an interesting direction would be a loss function which decreases the separation for more transferability.

I would like to increase my score to 5, and I strongly recommend the authors to incorporate these points into future versions.

**Time Spent Reviewing:**

6

---

> ### Author Response · Authors · 2021-08-10
> **Response to Reviewer Comments**
>
> We thank the reviewer for their positive comments regarding the importance of the problem, our writing, and the thoroughness of our experiments. However, it is not clear to us from the reviewer’s comments what specific improvements to the manuscript they would like to see or what weakness of the paper they believe merits rejection. We respond to each comment below.
>
> **Role of similarity between upstream and downstream tasks**
>
> >Although the conclusion of the paper is interesting and solid, I can be limited to the current setup. When the pretraning and target tasks are very similar (in extreme terms, the same), better performance on the pretraining task can lead to better transfer learning. As the similarity decreases, the performance of the pretraning and target tasks are less correlated, and we may enter the regime of the current paper.
>
> To summarize our experiments related to the effect of the downstream task: Across 8 different downstream natural image classification tasks, we demonstrate that higher class separation for the pretraining task is consistently associated with lower transfer accuracy. However, as we discuss in the paragraph starting on L224 and Section G of the Supplementary Material, when performing “transfer” of an ImageNet-pretrained model to a training set consisting of 40,000 examples from the ImageNet validation set, representations with higher class separation can be helpful rather than harmful.
>
> The 8 transfer tasks we investigate are commonly-used benchmark tasks for transfer learning in previous literature, and they more accurately represent real-world transfer scenarios than the ImageNet to ImageNet setting. In the typical transfer setting, the downstream task consists of different classes (or class-conditional distributions) than the pretraining task. The regime where we would expect greater class separation to lead to higher downstream accuracy is the regime where the downstream classes (and their distributions) are very similar to a subset of the pretraining classes, but in this regime, one could directly use the classifier from the pretraining task rather than transferring the representation, and thus it is not a typical transfer setting. We are happy to discuss this point in the paper’s limitations section.
>
> **Importance of findings regarding similarity of representations learned by different loss functions**
>
> >The findings of section 3.2 is not surprising as it is common sense that neural networks tend to learn similar “low-level” features in lower layers, and lower layers are more important in fine-tuning.
>
> The findings in Section 3.2 relate to differences between different loss functions. We agree that folk wisdom suggests that the first layer should learn Gabor-like filters regardless of the loss and the last layer should be specialized for the loss, but folk wisdom alone cannot tell us what happens in between. We do not think there is any previous work characterizing the layer at which the differences among different losses begin to appear, and we believe it is somewhat surprising that it is so late in the network (around block 13 out of 16 total ResNet blocks).
>
> >In addition, the accuracies of different objectives vary only by ~1%, which can be too small for an informative CKA visualization.
>
> Although accuracy differences between objectives are small, these small accuracy differences are associated with measurable differences in CKA in later blocks and in networks’ predictions (Supplementary Figure B.1), and thus we might reasonably expect to be able to detect differences in earlier layers with CKA.
>
> **Specificity of class separation to supervised pretraining**
>
> >As to the class separation, many pre-training tasks are not supervised classification, which do not involve it. Even if greater class separation is associated with less transferable features, it is hard to say class separation determines transferability, which cast doubt on the value of this claim.
>
> We accept that our findings are limited to pretraining on supervised image classification tasks, and we will add discussion of this to the paper’s limitations section. However, supervised image classification remains the most common paradigm for pretraining in computer vision, and thus, even with this limitation, our findings have important implications for real-world settings.

---

> > ### Author Response · Authors · 2021-08-31
> > **Please let us know if we have addressed your concerns**
> >
> > Dear Reviewer,
> >
> > Thank you for your helpful comments on our work. In our response, we believe we have replied to your main points. In the camera-ready submission, we plan to add additional clarifications regarding the setting in which our work applies and the interpretation of the findings in Section 3.2, which we believe will address your concerns. We would greatly appreciate it if you could update your review given our response.
> >
> > Thanks again for your time!

---

### Official Review · Reviewer_HjSp · 2021-07-20

**Rating:** 6
**Confidence:** 3

**Summary:**

This paper studies an interesting problem of how the representation features learned by different loss functions affect the performance on downstream tasks. They reveal some interesting patterns, such as better class separation on pre-training dataset does not guarantee better performance on downstream tasks and it can be caused by “overfitting” on original task. Overall, this paper is clearly written and easy to follow. The numerical results are also well-presented.

**Ethical Concerns:**




**Limitations And Societal Impact:**

This has been addressed in the paper.

**Main Review:**

Pros:
1) The authors present detailed studies on the choice of loss functions, comparison of transferability, etc., and draw several interesting conclusions.
2) Overall, I find this paper is clear and well-presented.

Cons:
1) ResNet is a sophisticated architecture, however, it needs more evidence to draw general conclusions about transferability, especially for section 3.2. For example, if we don’t use the residual connection, can we observe similar patterns?

2) The choice of downstream tasks is somewhat limited. To better demonstrate the transferability, different types of downstream tasks should be considered, e.g., object detection.

3) In line 224-230, the authors illustrate that representations with better class separation can perform better on downstream tasks if the classes are overlapped. So, I wonder how many overlapped classes are in these 8 selected datasets.

**Time Spent Reviewing:**

5

---

> ### Author Response · Authors · 2021-08-10
> **Response to Reviewer Comments**
>
> We thank the reviewer for their constructive comments. We address each below.
>
> **Role of architecture**
>
> >ResNet is a sophisticated architecture, however, it needs more evidence to draw general conclusions about transferability, especially for section 3.2. For example, if we don’t use the residual connection, can we observe similar patterns?
>
> We have performed similar experiments with Inception v3, which has no residual connections. For each loss, we trained 3 models, using the same loss function hyperparameters as for ResNet-50. We cannot include images in our response and thus cannot show the representational similarity plots here, but they are qualitatively similar to those shown in the paper. The patterns we observe in the transfer accuracies as well as the penultimate layer class separation are consistent with those for ResNet-50. We present the results in a table below. As in the paper, results not statistically significantly different from the best are bolded.
>
> |                     | ImageNet Top-1 | Top-5      | Class Separation ($R^2$) | Food   | CIFAR10   | CIFAR100   | Birdsnap   | SUN397   | Cars   | Pets   | Flowers   |
> |:--------------------|:---------------|:-----------|:-------------------------|:-------|:----------|:-----------|:-----------|:---------|:-------|:-------|:----------|
> | Softmax             | 78.6           | 94.24      | 0.356                    | **74.5** | **92.4**    | **76.2**     | **59.3**     | **63.1**   | **64.4** | 92.2   | **94.0**    |
> | Label Smoothing     | 78.8           | **94.60**  | 0.441                    | 73.3   | 91.6      | **75.0**     | 56.1       | 62.4     | 60.3   | **93.0** | 92.4      |
> | Sigmoid             | **79.1**       | 94.17      | 0.444                    | 73.7   | 91.3      | 74.7       | 55.0       | 62.0     | 60.7   | **92.8** | *93.0*    |
> | More Final Layer L2 | **79.0**       | 94.52      | 0.586                    | 70.1   | 91.0      | 73.3       | 52.4       | 61.0     | 51.1   | **92.5** | 89.6      |
> | Dropout             | **79.0**       | **94.50**  | 0.454                    | 72.6   | 91.5      | 74.7       | 56.3       | 62.1     | 59.2   | **92.7** | 92.2      |
> | Logit Penalty       | **78.9**       | **94.63**  | 0.638                    | 69.1   | 90.6      | 72.1       | 49.3       | 59.2     | 52.3   | 92.3   | 87.9      |
> | Logit Normalization | **78.8**       | 94.34      | 0.559                    | 67.4   | 90.6      | 72.2       | 50.9       | 58.5     | 45.6   | 92.1   | 84.2      |
> | Cosine Softmax      | **78.9**       | 94.38      | 0.666                    | 63.1   | 90.3      | 71.5       | 45.8       | 55.6     | 38.0   | 90.6   | 75.2      |
> | Squared Error       | 77.7           | 93.28      | 0.838                    | 45.3   | 84.1      | 57.6       | 25.0       | 41.1     | 18.8   | 85.7   | 54.8      |
>
> **Different types of downstream task**
>
> >To better demonstrate the transferability, different types of downstream tasks should be considered, e.g., object detection.
>
> We acknowledge that, in the current study, we have only examined transfer from one classification task to another. Although both upstream and downstream tasks are the same type of task, we nonetheless observe that some loss functions lead to representations that are over-specialized for the upstream task and perform worse for the downstream tasks. We expect that the detrimental impact of over-specialized representations would be even larger for downstream tasks with greater differences from the upstream task, and thus we do not expect that object detection tasks would yield different conclusions than the experiments we have already performed. However, if the reviewer feels that such experiments are important, we will consider adding them to the paper.
>
> **Class overlap between downstream tasks and ImageNet**
>
> >In line 224-230, the authors illustrate that representations with better class separation can perform better on downstream tasks if the classes are overlapped. So, I wonder how many overlapped classes are in these 8 selected datasets.
>
> Computing the amount of class overlap is difficult because of differences in granularity between ImageNet and other tasks. For example, CIFAR-10 contains a single “dog” class that corresponds to 90 dog breeds contained in ImageNet, but ImageNet contains only a single “hummingbird” class whereas Birdsnap contains 9 different species. To determine the numbers in the table below, we consider classes as “overlapping” when the name of the downstream either directly or nearly corresponds to an ImageNet class, or is a superclass of ImageNet classes.
>
> In addition to attempting to match the classes via their names, we also solve a variant of the assignment problem to map classes in the downstream dataset to ImageNet classes such that each ImageNet class maps to a single matching downstream class but a downstream class can map to several ImageNet classes. We find the mapping that maximizes accuracy of a softmax ResNet-50 model on the downstream training set and report accuracy on the downstream test set as "Assignment Acc." in the table below.
>
> | Dataset     | # Classes   | # Overlapping | Assignment Acc. |
> |:------------|:------------|:----------|:----------------|
> | Food        | 101         | 7         | 15.4%           |
> | CIFAR-10    | 10          | 9         | 65.1%           |
> | CIFAR-100   | 100         | 61        | 34.6%           |
> | Birdsnap    | 500         | 11        | 7.0%            |
> | SUN397      | 397         | 39        | 20.2%           |
> | Cars        | 196         | 0         | 4.0%            |
> | Pets        | 37          | 25        | 71.2%           |
> | Flowers     | 102         | 1         | 7.6%            |
>
> If one judges solely based on class names, CIFAR-10, CIFAR-100, and Pets all have substantial class-level overlap with ImageNet. However, the original classifier weights often fail to classify CIFAR-10 and CIFAR-100 images into corresponding ImageNet classes, likely due to the difference in resolution vs. the ImageNet training set. When we are constrained to express each class in the downstream dataset as a superset of ImageNet classes, we observe the highest accuracy on Pets.

---

> > ### Comment · Reviewer_HjSp · 2021-08-30
> > **Post-rebuttal Comments**
> >
> > I thank the authors for their detailed responses and the additional numerical experiments. Most of my concerns have been addressed. Overall, I think the quality of this paper is good, the results are convincing and well-presented. In addition, I still recommend the authors to try other complex networks.

---

### Decision · Program_Chairs · 2021-09-27

**Decision:**

Accept (Poster)

**Comment:**

This paper performed an empirical study on the impact of loss functions to the transferability of deep features for downstream tasks, and pinpointed several interesting findings which are previously unknown. Reviewers generally believed that this study is novel and significant to the community, the experimentation with several quantitative measures is solid, and the findings are conclusive and inspiring. Reviewers also raised some relevant concerns on the limited scope of this study, such as the limitation to ImageNet pre-training and less interesting downstream datasets, and to vision classification backbones and SGD optimizers. While these concerns are way of exhaustivity, authors are encouraged to extend the study to at least more interesting downstream datasets (such as Chest X-ray in the transfusion work, NeurIPS 2019) embodying a larger diversity of domain shifts, because this is the most relevant perspective to make their findings more conclusive for practical problems.